# Modulation of plant root growth by nitrogen source-defined regulation of polar auxin transport

Krisztina Ötvös[1,2] (iD), Marco Marconi[3] (iD), Andrea Vega[4] (iD), Jose O'Brien[4], Alexander Johnson[1] (iD), Rashed Abualia[1], Livio Antonielli[2], Juan Carlos Montesinos[1] (iD), Yuzhou Zhang[1] (iD), Shutang Tan[1] (iD), Candela Cuesta[1], Christina Artner[1], Eleonore Bouguyon[5] (iD), Alain Gojon[5], Jirí Friml[1], Rodrigo A. Gutiérrez[4] (iD), Krzysztof Wabnik[3,*] (iD) & Eva Benková[1,**] (iD)

## Abstract

Availability of the essential macronutrient nitrogen in soil plays a critical role in plant growth, development, and impacts agricultural productivity. Plants have evolved different strategies for sensing and responding to heterogeneous nitrogen distribution. Modulation of root system architecture, including primary root growth and branching, is among the most essential plant adaptions to ensure adequate nitrogen acquisition. However, the immediate molecular pathways coordinating the adjustment of root growth in response to distinct nitrogen sources, such as nitrate or ammonium, are poorly understood. Here, we show that growth as manifested by cell division and elongation is synchronized by coordinated auxin flux between two adjacent outer tissue layers of the root. This coordination is achieved by nitrate-dependent dephosphorylation of the PIN2 auxin efflux carrier at a previously uncharacterized phosphorylation site, leading to subsequent PIN2 lateralization and thereby regulating auxin flow between adjacent tissues. A dynamic computer model based on our experimental data successfully recapitulates experimental observations. Our study provides mechanistic insights broadening our understanding of root growth mechanisms in dynamic environments.

Keywords auxin transport; nutrients; post-translational modification; protein trafficking; root development
Subject Categories Development; Membranes & Trafficking; Plant Biology
The EMBO Journal (2021) 40: e106862

## Introduction

The ability to sense and adapt to fluctuations in nutrient availability is essential for the survival of all organisms. Every life form on our planet possesses delicate mechanisms for sensing and reacting to the variable nutrient status and adjusts their behavior to maintain growth or cope with stress caused by malnutrition. Mineral nutrients absorbed from the soil are major determinants of plant growth and development. Although required, fluctuations in their availabilities either to sub- or supra-optimal levels often have detrimental effects on plant metabolism and physiology, thereby attenuating plant fitness. Hence, the acquisition of mineral nutrients from the soil needs to be tightly controlled and endogenous levels within a plant body maintained at a physiological optimum level. At the molecular level, balancing nutrient acquisition with the plant's requirements implies that there is close communication between pathways controlling uptake, distribution and homeostasis of nutrients, and the pathways coordinating plant growth and development.

The root system perceives and integrates local and systemic signals on the nutrient status to regulate activity of pathways mediating nutrient uptake and distribution. An important component of the plant's nutrient management strategy involves a rapid modulation of the root growth and development. In response to nutrient availability, root meristem activity and elongation growth of primary root, as well as root branching, are adjusted in order to optimize nutrient provision to the plant body (López-Bucio et al, 2003). Production of new cells is essential for sustainable root growth; however, enhancement of the cell division machinery typically occurs within a range of hours (Marhava et al, 2019). In contrast, rapid modulation of cell elongation and manifold increase in cell volume would ensure faster growth responses (Fendrych et al, 2016). Hence, in fluctuating environmental conditions root

1   Institute of Science and Technology (IST) Austria, Klosterneuburg, Austria
2   Bioresources Unit, Center for Health & Bioresources, AIT Austrian Institute of Technology GmbH, Tulln, Austria
3   Centro de Biotecnología y Genómica de Plantas (CBGP, UPM-INIA) Universidad Politécnica de Madrid (UPM)—Instituto Nacional de Investigación y Tecnología Agraria y Alimentaria (INIA), Madrid, Spain
4   Pontifical Catholic University of Chile, Santiago, Chile
5   BPMP, CNRS, INRAE, Institut Agro, Univ Montpellier, Montpellier, France
    *Corresponding author. Tel: +34 910679203; E-mail: k.wabnik@upm.es
    **Corresponding author. Tel: +43 2243 9000 5301; E-mail: eva.benkova@ist.ac.at

growth kinetics relies on the coordination of rapid elongation growth and adjustment of proliferation activity of the meristem.

Nitrogen (N) is a key macronutrient present in many key biological molecules and therefore constitutes a limiting factor in agricultural systems (Marschner, 2008). Although plants are dependent on an exogenous N supply and use nitrate ($NO_3^-$), nitrite ($NO_2^-$), and ammonium ($NH_4^+$) as major sources of inorganic N, their preference for different inorganic forms depends on plant adaptation to soil (von Wirén et al, 1997; Marschner, 2008). For example, wheat, maize, canola, beans, sugar beet, Arabidopsis, and tobacco grow preferentially on $NO_3^-$ nutrition, whereas rice and pine grow on $NH_4^+$ nutrition. Fluctuations in both concentrations and the form of nitrogen sources available in the soil have prominent effects on root system growth and development (Jia & Wirén, 2020; Waidmann et al, 2020). Deficiency in nitrogen severely interferes with root elongation growth and development; low to medium availability of nitrogen enhances root growth and branching to promote the exploitation of this macronutrient, whereas high levels of availability might inhibit the elongation growth of primary and lateral roots (Gruber et al, 2013). When exposed to local nitrate-rich zones, the root system responds by enhancing lateral root (LR) outgrowth (Remans et al, 2006; Forde, 2014; Giehl & Wirén, 2014). In the model plant Arabidopsis thaliana, the local availability of $NO_3^-$ and $NH_4^+$ seems to have complementary effects on the LR development [$NH_4^+$ stimulates branching, whereas $NO_3^-$ induces LR elongation (Remans et al, 2006; Lima et al, 2010)]. These complex adaptive responses of the root organ to N sources and heterogeneity in availability are regulated by a combination of systemic and local signaling (Fredes et al, 2019). The impact of available sources of N on the root system is closely interconnected with the activity of plant hormones including auxin, cytokinin, ABA, ethylene, and others (Ristova et al, 2016; Krouk, 2016; Guan, 2017). In recent years, a number of studies have demonstrated that auxin biosynthesis, transport, and accumulation are altered in response to different N regimes in maize (Tian et al, 2008; Chen et al, 2013), soybean (Caba et al, 2000), pineapple (Tamaki & Mercier, 2007), and Arabidopsis thaliana (Walch-Liu et al, 2006; Krouk et al, 2010; Ma et al, 2014; Krouk, 2016). In Arabidopsis, several key auxin-related regulatory modules that respond to nitrogen availability were identified including TAR2, a gene involved in auxin biosynthesis, transporters of auxin such as PIN-FORMED 1 (PIN1), PIN2, PIN4, and PIN7 and molecular components, which control their subcellular trafficking (Gutiérrez et al, 2007; Krouk et al, 2010). At the level of auxin signaling, Auxin Response Factor AUXIN RESPONSE FACTOR 8 (ARF8, encoding a transcription factor of the auxin signaling machinery) was identified as a N responsive gene in the pericycle (Gifford et al, 2008). ARF8 together with its associated microRNA167s is involved in the control of the ratio between LR initiation and emergence (Gifford et al, 2008; Vidal et al, 2010, 2013a, 2013b). Another mechanism of nitrogen–auxin interplay underlying adaptation of the root system is mediated through NRT1.1, the nitrate transceptor (Tsay et al, 1993). Its dual auxin–nitrate transport activity has been shown to play an important role in the adaptation of the root system, in particular, LR emergence to nitrate availability (Krouk et al, 2010; Mounier et al, 2014).

Flexible modulation of primary root growth to fluctuations in nitrogen resources has been recognized as a prominent foraging strategy to optimize N exploitation (Gifford et al, 2013). However, the mechanisms that control the rapid reconfiguration of root growth dynamics in response to diverse N sources are still poorly understood. Here, to dissect the tissue and cellular mechanisms underlying the early phases of this adaptive process we focused on the primary responses of Arabidopsis roots to alterations in the available source of N such as $NH_4^+$ and $NO_3^-$. We performed real-time vertical confocal imaging to capture the earliest root responses after the replacement of $NH_4^+$ by $NO_3^-$. We found that in roots supplied with $NH_4^+$, local attenuation of meristematic activity in the epidermis results in the earlier transition of epidermal cells into elongation when compared to the cortex, thus generating asynchronous elongation of the adjacent tissues. Substitution of $NH_4^+$ for $NO_3^-$ led to a rapid enhancement of root growth associated with the simultaneous entrance of more cells at the root transition zone into elongation, and the subsequent re-establishment of a critical balance between cell proliferation and elongation in the adjacent cortex and epidermis. We demonstrate that epidermis and cortex tissues require $NO_3^-$ for synchronous growth. We show that the essential mechanism underlying this flexible adaptation of root growth involves nitrate-dependent regulation of the auxin transport. In roots supplied with different forms of N, distinct localization patterns of the PIN2 auxin efflux carrier are generated as a result of dynamic PIN2 subcellular trafficking. Intriguingly, phosphoproteome analysis of PIN2 (Vega et al, 2020) led to the identification of a previously uncharacterized nitrate-sensitive phosphorylation site. The functional characterization of PIN2 and its phospho-variants suggest that the N source-dependent modulation of PIN2 phosphorylation status has a direct impact on the flexible adjustment of PIN2 localization pattern and thereby facilitates the adaptation of root growth to varying forms of N supply. Finally, we integrated experimental data regarding the nitrogen-dependent root growth into a quantitative computer model. Our computer model recapitulated in planta patterning from a minimal set of assumptions and made predictions that were tested experimentally. Taken together, we present a quantitative, mechanistic model of how Arabidopsis primary root growth is fine-tuned to different N sources through growth synchronization of adjacent tissues. We hypothesize that the flexible modulation of growth patterns relying on nutrient response on auxin transport is an important part of the successful strategy that enables plant root adaption to the dynamically changing environment and thus maintains its sustainable growth.

## Results

### Root growth rapidly adjusts to form of nitrogen source

To explore how primary root responds and adapts to different forms of N, Arabidopsis seedlings were grown on $NH_4^+$ as an exclusive N source for 5 days (5 DAG) and afterward transferred on media containing either $NH_4^+$ or $NO_3^-$. We found that replacement of $NH_4^+$ by $NO_3^-$ rapidly enhanced root length and already 6 h after transfer (HAT), roots were significantly longer compared to those supplied with $NH_4^+$ (Fig EV1A). In general, root growth is determined by the elongation of cells, which are constantly produced by the root apical meristem. To study processes that underlie the adaptation of root growth to different forms of N, a vertical confocal microscope

equipped with a root tracker system was employed. Using this setup, we were able to detect and monitor the earliest root responses with a high cellular resolution (von Wangenheim *et al*, 2017). To minimize the interference of physiological conditions for seedling development, a light–dark regime was maintained in course of the root tracking. After the transfer of wild-type (Col-0) seedlings to $NH_4^+$ containing medium root growth rate (RGR) was enhanced, presumably as a response to stress caused by transfer of seedlings to a fresh plate. Within ~120 min, RGR stabilized at an average speed of $1.37 \pm 0.025$ μm/min. Transition to dark period correlated with a rapid drop of RGR to $0.98 \pm 0.029$ μm/min, which was maintained during the dark phase and at the light recovered again to $1.27 \pm 0.048$ μm/min. Seedlings transferred to $NO_3^-$ reacted by an increase of RGR to $1.77 \pm 0.042$ μm/min and similarly to roots on $NH_4^+$, during the dark period their RGR decelerated and was retrieved to $1.81 \pm 0.051$ μm/min at the light (Fig 1A, Movie EV1). Hence, provision of $NO_3^-$ caused a rapid enhancement of RGR when compared to $NH_4^+$, but it did not interfere with its circadian rhythmicity (Yazdanbakhsh *et al*, 2011).

To gain more insight into the mechanistic basis underlying the rapid increase of root length after substitution of $NH_4^+$ for $NO_3^-$, we focused on cells in the transition zone (TZ). The TZ is located between the root apical meristem and elongation zone, and cells while passing this developmental zone undergo essential modifications associated with their transition from the proliferative to the elongation phase (Baluska *et al*, 2010; Kong *et al*, 2018) (Fig 1B). Time-lapse experiments capturing root growth from 2 to 3.76 h after transfer combined with a tracking of cell membranes pointed at differences in the elongation pattern of epidermal cells in roots supplied with either $NH_4^+$ or $NO_3^-$, while in roots supplemented with $NH_4^+$ only a few epidermal cells enter into elongation phase. Provision of $NO_3^-$ increased number of elongating cells in the TZ (Fig 1B, Movie EV2). Next, we analyzed in detail 18 roots 12 HAT on either $NH_4^+$ or $NO_3^-$ and measured length of the epidermal cells across the meristematic, transition, and the start of the elongation zones. The analyses suggested that on $NO_3^-$ more epidermal cells enter into transition phase, as indicated by an increased number of cells 30–40 μm long when compared to roots on $NH_4^+$ (Fig EV1B). Despite the stimulating impact of $NO_3^-$ on cell transition into the elongation phase, no differences in the maximal length of fully differentiated epidermal cells between roots on $NO_3^-$ and $NH_4^+$ were detected (Fig EV1C). This suggests that $NO_3^-$ promoted root growth is a result of modulated elongation kinetics of cells along the longitudinal root growth axis and not increase of the maximal cell length.

To sustain root growth, the rate of cell elongation and differentiation has to be tightly balanced with the production of new cells in the root meristem (Pavelescu *et al*, 2018). Hence, enhanced growth of cells after replacement of $NH_4^+$ by $NO_3^-$ could lead to depletion of the meristem if expansion of cells would prevail over a new cell production. To examine how root meristem adapts to change in N supply, cell length and frequency of divisions in epidermis and cortex along the longitudinal root growth axis were closely inspected 12 HAT. Surprisingly, length of epidermal cells started to increase from the 11[th] cell on (cell number was counted from quiescent center (QC)) in roots supplied with $NH_4^+$ (Fig 1C, and Appendix Fig S9A and B). In contrast, roots on $NO_3^-$ exhibited an increase in size from the 13[th] epidermal cells (Figs 1D and EV1D, Appendix Fig S9A

and B). Unlike the epidermis, the length profiles of cortex cells were not significantly different between roots supplied with either $NH_4^+$ or $NO_3^-$ (Fig EV1E, Appendix Fig S9A and B). Therefore, the growth of epidermal and cortex cells in roots on $NH_4^+$ displayed clearly asynchronous behavior whereas on $NO_3^-$ growth of these two adjacent cell files was synchronized (Fig 1C and D). Additionally, a machine learning approach was applied to regression analysis for assessing the importance of each variable (i.e., treatments: ammonium and nitrate, tissues: epidermis and cortex, cell positions) on cell length differences. Analysis of deviance was followed by estimated marginal mean (emmean) comparisons of cell lengths in different tissues (epidermis vs. cortex) at each cell position (1–20 from QC) for each treatment (ammonium vs. nitrate). The results show that ammonium and nitrate treatments affect the cell positions differentially: epidermal cells from the 17[th] up to the 20[th] position are significantly longer on ammonium while cell length in cortex is not affected by the treatments. (Appendix Fig S9A and B). Results were confirmed by recursive partitioning analysis and shown in a decision tree (Appendix Fig S9C).

The distinct elongation pattern of epidermal and cortex cells detected in roots on $NH_4^+$ can only be sustained if cell divisions in cortex compensate for an earlier start of cell elongation in the epidermis. Accordingly, scoring of cell division events (visualized by DAPI) revealed a higher number of mitotic events in cortex compared to epidermal cells in roots transferred to $NH_4^+$. On $NO_3^-$, similar frequency of cell divisions in both epidermal and cortex cell files was observed (Fig 1E, Appendix Fig S1A). Finally, monitoring of the cell cycle reporter *CyclinB::GUS* expression 2 days after transfer (DAT) to either $NH_4^+$ or $NO_3^-$ revealed enhanced reporter expression and overall enlargement of the meristematic zone in roots supplemented with $NO_3^-$ (Appendix Fig S1B).

Altogether, these data indicate that roots adopt distinct growth strategies involving fine-tuning of cell division and expansion across adjacent tissues to adapt to different forms of N. In roots supplied with $NH_4^+$, the meristematic activity of epidermal cells is attenuated, which results in their earlier transition into the elongation phase when compared to the cortex and asynchronous growth. Provision of $NO_3^-$ increases the number of epidermal cells in the TZ (Figs 1B and EV1B), which is one of the earliest detectable adaptive responses. Subsequently, within 12 h, the frequency of cell division in the epidermis increases, which results in shift of balance between cell division and elongation and more synchronized growth of cortex and epidermis. Eventually, a long-term supply of $NO_3^-$ enables enlargement of the root apical meristem compared to roots supplied with $NH_4^+$.

## Level and pattern of auxin activity in roots are modulated by form of nitrogen source

The plant hormone auxin is an essential endogenous regulatory cue that determines key aspects of root growth. Interference with auxin biosynthesis (Stepanova *et al*, 2008), signaling (Dello Ioio *et al*, 2008) or distribution (Blilou *et al*, 2005) at the root tip has a significant impact on the meristem maintenance, and transition of meristematic cells into elongation and differentiation phase. Distinct growth patterns observed in roots supplemented with different forms of N prompted us to monitor distribution of auxin at the root tip. Quantification of LUCIFERASE activity in protein extracts from

roots carrying the auxin sensitive *DR5::LUCIFERASE* reporter revealed that already 1 h after transfer to $NO_3^-$ containing medium the auxin response increases when compared to roots transferred to $NH_4^+$ supplemented medium (Appendix Fig S2A). To closely inspect the auxin distribution in a cell lineage-specific manner, a ratiometric degradation-based *R2D2* auxin reporter was implemented (Liao *et al*, 2015). In accordance with observations based on the *DR5::LUCIFERASE* reporter, a decreased ratio between DII-Venus (green) and mDII-Tomato (red) fluorescent signals indicated increased levels of auxin activity in the central cylinder of roots in response to replacement of $NH_4^+$ by $NO_3^-$ (Appendix Fig S2B).

In addition, we focused on the detailed profiling of the R2D2 reporter in the epidermis and the cortex (Appendix Fig S2C). Interestingly, we detected an overall increase of auxin activity in epidermal cells when compared to cortex cells in roots supplied with $NH_4^+$, whereas no difference between these two cell files was detected in roots on $NO_3^-$ (Fig 2A and B). Furthermore, on $NH_4^+$ there was an increase of auxin activity in epidermal cells when compared to cortex cells (starting from ~ 11[th] cell from the QC), while in roots supplied with $NO_3^-$ the auxin activity profiles followed similar trends of steady increase in both cortex and epidermal cell files (Fig 2A and B). Altogether, these analyses indicate that

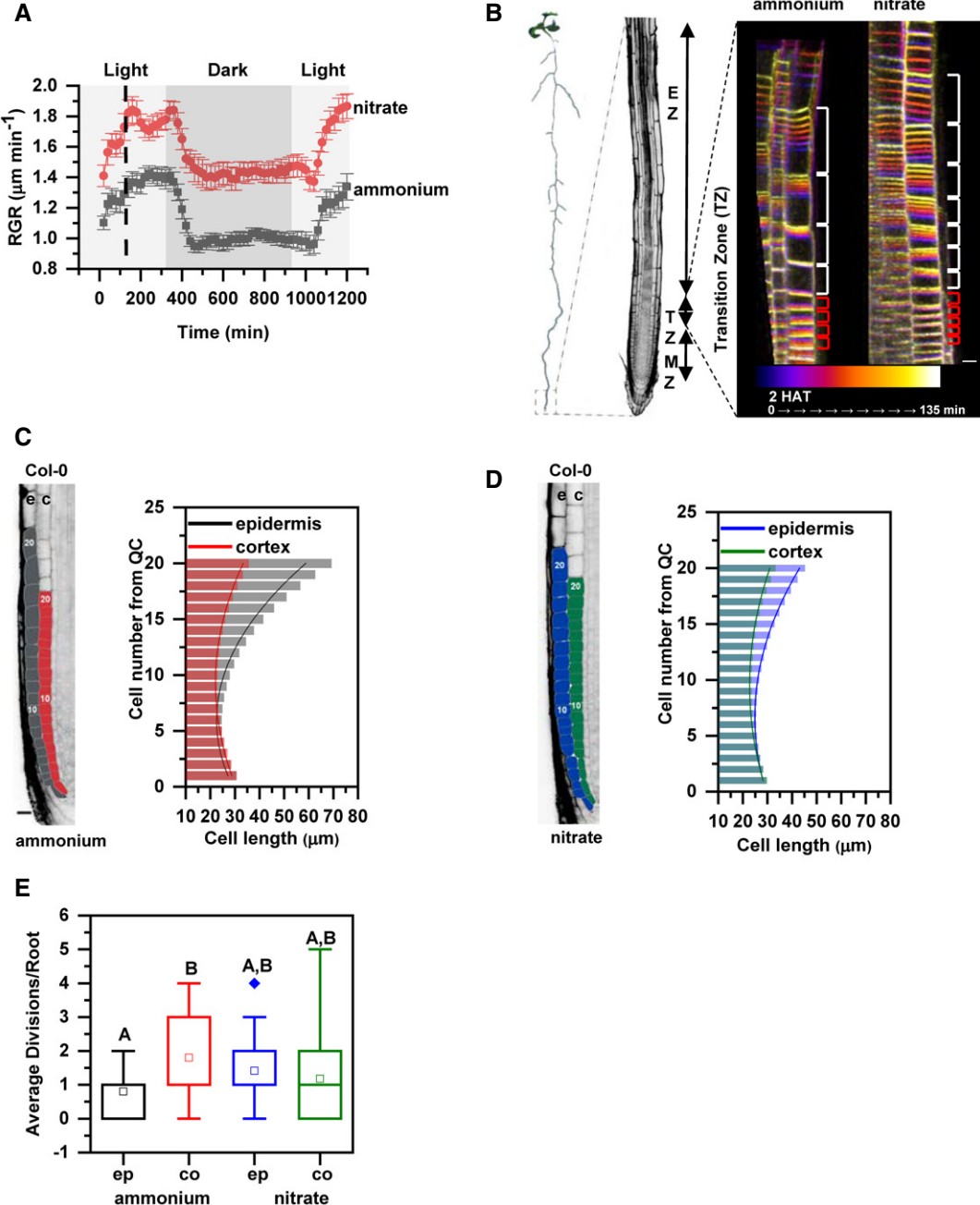

**Figure 1.**

**Figure 1.  Primary root growth kinetics of *Arabidopsis (Arabidopsis thaliana (L.) Heynh. Columbia-0,* Col-0) on ammonium or nitrate-containing medium.**

A       Seedlings were transferred 5 days after germination (DAG) to medium supplemented with ammonium (gray) or nitrate (red). Root growth rates (RGR in µm/min)
         were monitored over a 1,200 min period. Data represent the geometric mean ($\pm$ standard error, SE) of 3 biological replicates, $n = 5$ (ammonium) and 6 (nitrate)
         roots. Light and dark periods are indicated as light or dark gray background, respectively.

B       On the left, schematic representation of distinct root zones: Meristematic Zone (MZ), Transition Zone (TZ) and Elongation Zone (EZ). On the right, time-lapse
         imaging of cell growth at the TZ. Cells were visualized using the plasma membrane marker (wave line W131Y). Observation of roots started 2 h after transfer
         (2 HAT; blue) on ammonium or on nitrate for 135 min (white) and images were recorded every 20 min (9 stacks/root/recording). Red and white brackets indicate
         the length of meristematic and elongating cells at the last measurement point, respectively. Scale bar = 30 µm.

C, D   Representation and quantification of cell length in epidermal (e) and cortical (c) cell files. Optical, longitudinal sections of 5 DAG old Col-0 roots 12 HAT to
         ammonium (C) or nitrate (D) supplemented media. The first 20 epidermal (e) and cortex (c) cells (from quiescent center (QC)) are highlighted in gray and in red on
         ammonium (C), and in blue and green on nitrate (D), respectively. Scale bar = 30 µm. Column bars denote the geometric mean of the cell lengths at the respective
         positions. Lines represent a polynomial regression fit, with calculated slopes between cells 10 and 20 of $3.32639 \pm 0.17172$ (ammonium, epidermis),
         $1.22033 \pm 0.08754$ (ammonium, cortex) and $1.70502 \pm 0.09532$ (nitrate, epidermis), $0.82342 \pm 0.06973$ (nitrate, cortex). Data are derived from 3 biological
         replicates, $n = 18$ roots in each case.

E       Graphical representation of the average number of cell divisions along epidermis (ep) and cortex (co) in 5 DAG root tips 12 HAT to ammonium or nitrate
         supplemented media. Data are derived from 15 and 17 roots. The statistical significance was evaluated with ANOVA at $P < 0.05$. The box chart components are
         defined as; box (25–75%), central band (median line), central box (mean), and the range are within 1.5IQR.

Source data are available online for this figure.

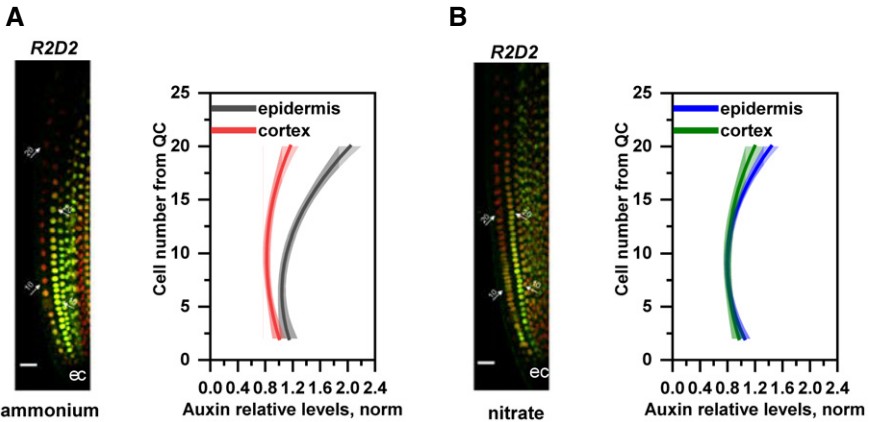

**Figure 2.  Relative auxin level in Col-0 root tips transferred to medium supplemented with ammonium or nitrate.**

A, B   Maximum intensity Z-stack projection images of 5 DAG old roots expressing the *R2D2* auxin signaling reporter 12 HAT to ammonium (A) or nitrate (B)
         supplemented media. White arrows mark the position of the $10^{th}$ and $20^{th}$ cells from QC; "e" and "c" mark epidermis and cortex, respectively. Scale bar = 50 µm.
         Graphs denote normalized relative auxin levels at the respective positions. Lines represent polynomial regression fit with 95% confidence band. Data are derived
         from 5 roots per condition from 3 biological replicates.

Source data are available online for this figure.

pattern of auxin activity at root meristems might adapt to specific N conditions. In roots supplied with $NH_4^+$, the early steep gradient of auxin signaling in epidermal cells correlates with their early transition into rapid elongation phase. Whereas in cortex cells, auxin reaches concentrations which might drive elongation in the more proximal cells. Substitution of $NH_4^+$ by $NO_3^-$ attenuates differences in profiles of auxin distribution between the cortex and the epidermal cell files, which would lead to synchronized cell growth (Fig 2A and B compared to Fig 1C and D).

**Nitrogen source affects basipetal auxin transport**

Directional cell-to-cell transport of auxin significantly contributes to the establishment of the auxin activity pattern at the root tip. The polar auxin transport (PAT) machinery, composed of AUX/LAX influx and PIN efflux carriers, directs the flow of auxin from the shoot acropetally through the stele toward the root tip; from where

it is via epidermis basipetally redistributed to the elongation zone. At the TZ, auxin might be redirected from the basipetal stream across the cortex, endodermis, and pericycle back to the stele and the root tip, thereby fine-tuning levels of auxin at the TZ (Goldsmith, 1977; Adamowski & Friml, 2015). The modulation of auxin activity pattern in the outer tissues detected after the replacement of $NH_4^+$ for $NO_3^-$ suggests that there are alterations of the basipetal auxin transport. To explore how different forms of N affect the flow of auxin in basipetal direction, transport assays using radioactively labeled auxin ($^3$H-IAA) were performed. Six hours after applying $^3$H-IAA to the root tip, radioactivity in the proximal zone of the primary roots supplied with $NH_4^+$ was significantly lower when compared to roots on either $NO_3^-$ supplemented or standard Murashige and Skoog (MS) medium (Fig EV2A). These results indicate that basipetal auxin transport can be modulated by available source of N, and provision of $NO_3^-$ enhances flux of auxin in shootward direction when compared to $NH_4^+$.

                                           

The PIN2 auxin efflux carrier is among the principal components of PAT mediating basipetal transport of auxin in roots (Luschnig *et al*, 1998; Müller *et al*, 1998). To test whether adjustment of the basipetal auxin flow in response to different sources of nitrogen is dependent on activity of PIN2, we tested *eir1-4*, a mutant defective in this efflux transporter. In agreement with previous reports (Hanzawa *et al*, 2013), a significantly lower radioactivity in the proximal root zone of the *eir1-4* was detected when compared to wild-type roots on MS medium (Fig EV2A). Noteworthy, no radioactivity increase in the proximal zone of *eir1-4* roots was observed in roots supplied with $NO_3^-$ when compared to $NH_4^+$ (Fig EV2A), pointing toward PIN2 function in the flexible adjustment of the basipetal auxin flow in response to form of N source. To further examine the role of the PIN2 mediated transport in establishment of distinct auxin patterns at root tips supplemented with different forms of N, we monitored the auxin sensitive reporter DII-Venus and its stabilized auxin-responsive analog mDII-Venus (Brunoud *et al*, 2012) as a reference in *eir1-4* and Col-0 roots. The expression pattern of DII-Venus reporter in Col-0 roots was largely consistent with what we observed using the R2D2 reporter. In Col-0 roots supplied with $NH_4^+$, a reduced DII-Venus signal indicated a higher auxin activity in epidermal cells when compared to the cortex. Also, consistently with the R2D2 reporter, a steeper slope of auxin activity in epidermis when compared to cortex (with onset at ~8[th] cell distance from QC) was detected in roots supplied with $NH_4^+$, whereas in roots on $NO_3^-$, auxin activity both in epidermis and cortex followed similar trends (Fig EV2B and C compared to Fig 2A and B and Appendix Fig 3A and B). Conversely, *eir1-4* was severely affected in adjustment of auxin pattern to different N sources. When compared to Col-0, overall higher levels of auxin activity in both epidermal and cortex cells and a shallower slope of auxin activity increase in the epidermis was observed in *eir1-4* roots supplied with $NH_4^+$. As a result, the difference in auxin activity profiles between the cortex and the epidermis in *eir1-4* was less pronounced than in wild-type roots (Fig EV2D compared to Fig EV2B and Appendix Fig S3A). On $NO_3^-$, the overall profiles of auxin activity in epidermis and cortex of *eir1-4* followed similar trends, characterized by shallow slope along the longitudinal root growth axis (Fig EV2E, Appendix Fig S3B). Importantly, expression pattern of the auxin insensitive mDII-Venus reference construct remained largely unchanged under all tested conditions in both wild type and *eir1-4* (Appendix Fig S3C and D). Altogether, our results point at an important role of PIN2 dependent basipetal auxin transport in adjustment of auxin activity pattern in roots to specific N conditions.

### PIN2 mediates root growth adaptation to nitrogen resources

To further examine the role of PIN2-mediated basipetal auxin transport in root growth adaptation to different sources of N, *eir1-4* and *eir1-1*-mutant alleles of *PIN2* were analyzed. Unlike in wild type, no significant increase in root length was detected 1 DAT in either *eir1-4* or *eir1-1* seedlings on $NO_3^-$ when compared to $NH_4^+$ supplemented medium (Appendix Fig S4A). Closer inspection of the RGR in real time using the vertical confocal–root tracking set up showed that after transfer on $NH_4^+$ growth of the *eir1-4* roots stabilized at $1.47 \pm 0.041$ μm/min and $1.35 \pm$ μm/min during light and dark period, respectively. However, no significant increase of RGR after

transfer to $NO_3^-$ containing medium could be observed (Fig 3A). These results strongly support an essential role of PIN2-mediated basipetal auxin transport in rapid adjustment of root growth to form of nitrogen source.

To explore whether *eir1-4* root growth adapts to different forms of N, elongation patterns of epidermal and cortex cells were analyzed. Measurements of cell lengths along the longitudinal growth axis of *eir1-4* roots supplied with $NH_4^+$ revealed that unlike in Col-0, epidermal cells undergo gradual, steady elongation growth comparable to that in cortex. Notably, patterns of cortex and epidermal cell growth in *eir1-4* appear more synchronous than in wild-type roots on $NH_4^+$ (Fig 3B vs. Fig 1C). In *eir1-4* roots 12 HAT from $NH_4^+$ to $NO_3^-$ supplemented medium, we observed largely synchronized pattern of elongation in both epidermal and cortex cell files, characterized by gradual, steady increase of cell length similar to these observed in Col-0 (Figs 3C and 1D). Consistently with a more synchronous pattern of epidermal and cortex cell growth in both N regimes, no significant differences in frequency of mitotic events between epidermis and cortex were found in *eir1-4* roots on medium supplied with either $NH_4^+$ or $NO_3^-$ (Fig 3D).

Overall, loss of PIN2 activity interfered with enhancement of root growth in response to $NO_3^-$ provision and affected the establishment of tissue-specific growth patterns typically adopted by Col-0 roots supplied with different sources of N. Altogether, these results indicate that PIN2-mediated basipetal and lateral auxin transport plays an important function in acquiring distinct root growth patterns during adaptation to different N sources.

### PIN2 delivery to the plasma membrane and polarity is adjusted in response to form of nitrogen source

To explore the mechanisms underlying PIN2 function in root growth adaptation to different N sources, we examined its expression, abundance at the plasma membrane (PM), and subcellular trafficking in roots supplied with $NH_4^+$ or $NO_3^-$. RT–qPCR analyses of 7 DAG roots grown on $NH_4^+$ and transferred to media supplemented with either $NH_4^+$ or $NO_3^-$ for 1, 6, and 48 h did not reveal any significant changes in *PIN2* transcription in any of the tested conditions (Fig EV3A). Likewise, expression of neither the *PIN2::nlsGFP* nor the *PIN2::GUS* reporter was affected by different N source (Fig EV3B). Interestingly, monitoring of *PIN2::PIN2-GFP* transgenic seedlings revealed significantly increased abundance of the PM-located PIN2-GFP in epidermal and cortex cells of roots supplied with $NO_3^-$ when compared to $NH_4^+$ (Fig 4A). Furthermore, in cortex cells at the transition zone of $NO_3^-$ supplied roots, besides expected localization at the apical PM (Blilou *et al*, 2005), enhanced lateralization of PIN2-GFP to the inner and outer PMs could be detected (Figs 4B and EV3D). Immunolocalization using PIN2-specific antibodies is fully consistent with the observations of PIN2-GFP and ruled out possible interference with fluorescence of GFP reporter by different N source (Appendix Fig S5A–C). Hence, substitution of $NH_4^+$ by $NO_3^-$ seems to affect PIN2 at the post-transcriptional rather than at the transcriptional level.

PIN proteins constantly recycle between the PM and endosomal compartments, and thus, their abundance at the PM is largely dependent on a balance between endo- and exocytosis (Kleine-Vehn & Friml, 2008; Kleine-Vehn *et al*, 2011). Hence, we explored whether modulation of PIN2 subcellular trafficking is the

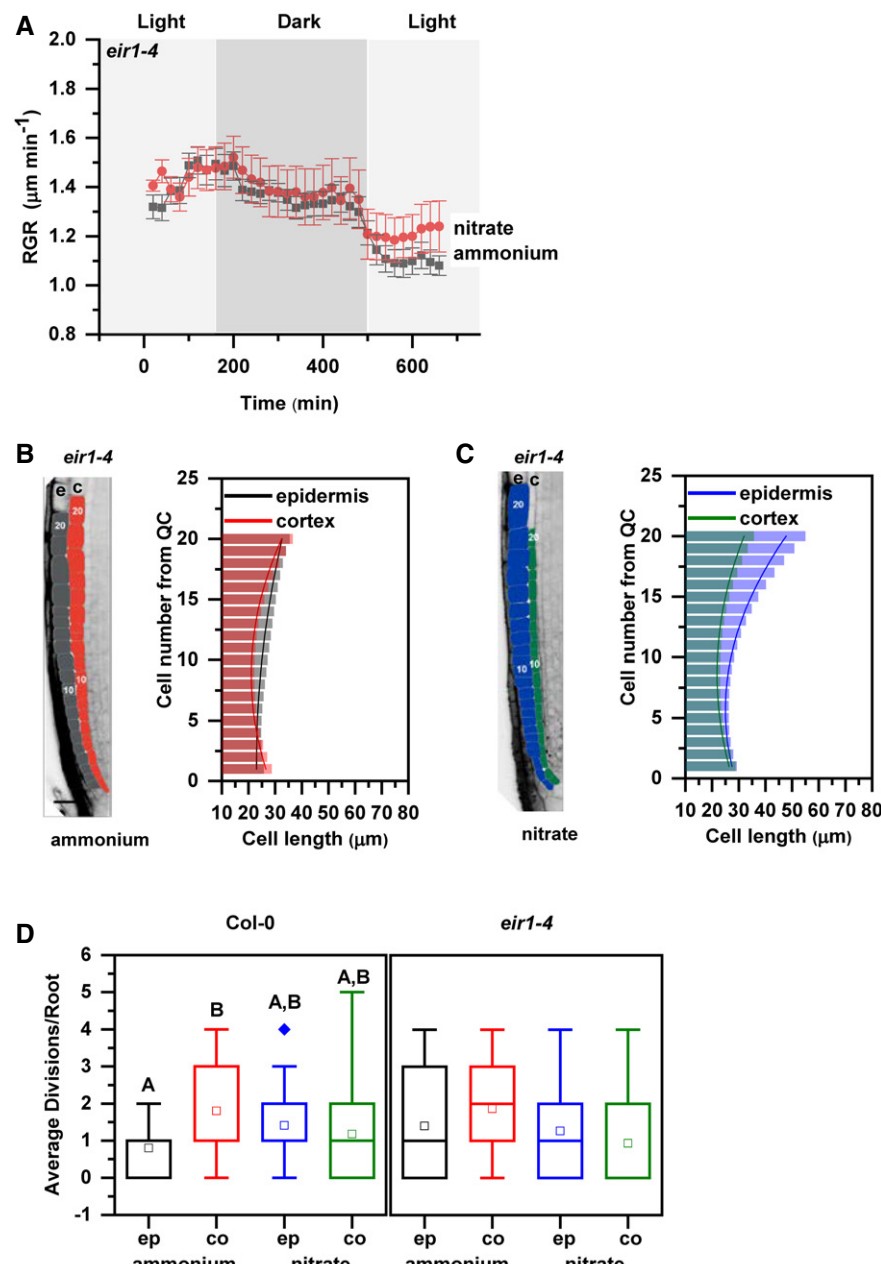

**Figure 3. Primary root growth kinetics of *eir1-4* roots transferred to ammonium or nitrate amended medium.**

A   Root growth rate (RGR in µm/min) of *eir1-4* roots transferred 5 DAG to ammonium (gray) or nitrate (red) containing medium over a period of 680 min. Data represent the geometric mean (± standard error, SE) of 3 biological replicates (*n* = 5 roots/condition). Light and dark periods are highlighted in light or dark gray.

B, C   Representation and quantification of cell length in epidermal (e) and cortical (c) cell files. Optical, longitudinal sections of 5 DAG *eir1-4* roots 12 HAT to ammonium (B) or nitrate (C) supplemented media. The first 20-20 epidermal and cortex cells (from quiescent center (QC)) are highlighted in gray and in red on ammonium (B) and in blue and green on nitrate (C), respectively. Scale bar = 30 µm. Column bars denote the geometric mean of cell length at the respective positions. Lines represent a polynomial regression fit, with calculated slopes between cells 10 and 20 of 0.75884 ± 0.02624 (ammonium, epidermis), 1.13088 ± 0.08446 (ammonium, cortex) and 2.06912 ± 0.10341 (nitrate, epidermis), 0.99878 ± 0.07278 (nitrate, cortex). Data are derived from 3 biological replicates, *n* = 9 (ammonium) and 8 (nitrate) roots.

D   Average number of cell divisions along the epidermis (ep) and cortex (co) in 5 DAG old Col-0 and *eir1-4* root tips 12 HAT to ammonium or nitrate supplemented media. Data are derived from *n* = 15 and *n* = 17 roots of Col and *n* = 10 and *n* = 9 roots of *eir1-4* on ammonium and nitrate, respectively. Statistical significance was evaluated with ANOVA at *P* < 0.05. The box chart components are defined as; box (25–75%), central band (median line), central box (mean), and the range is within 1.5IQR.

Source data are available online for this figure.

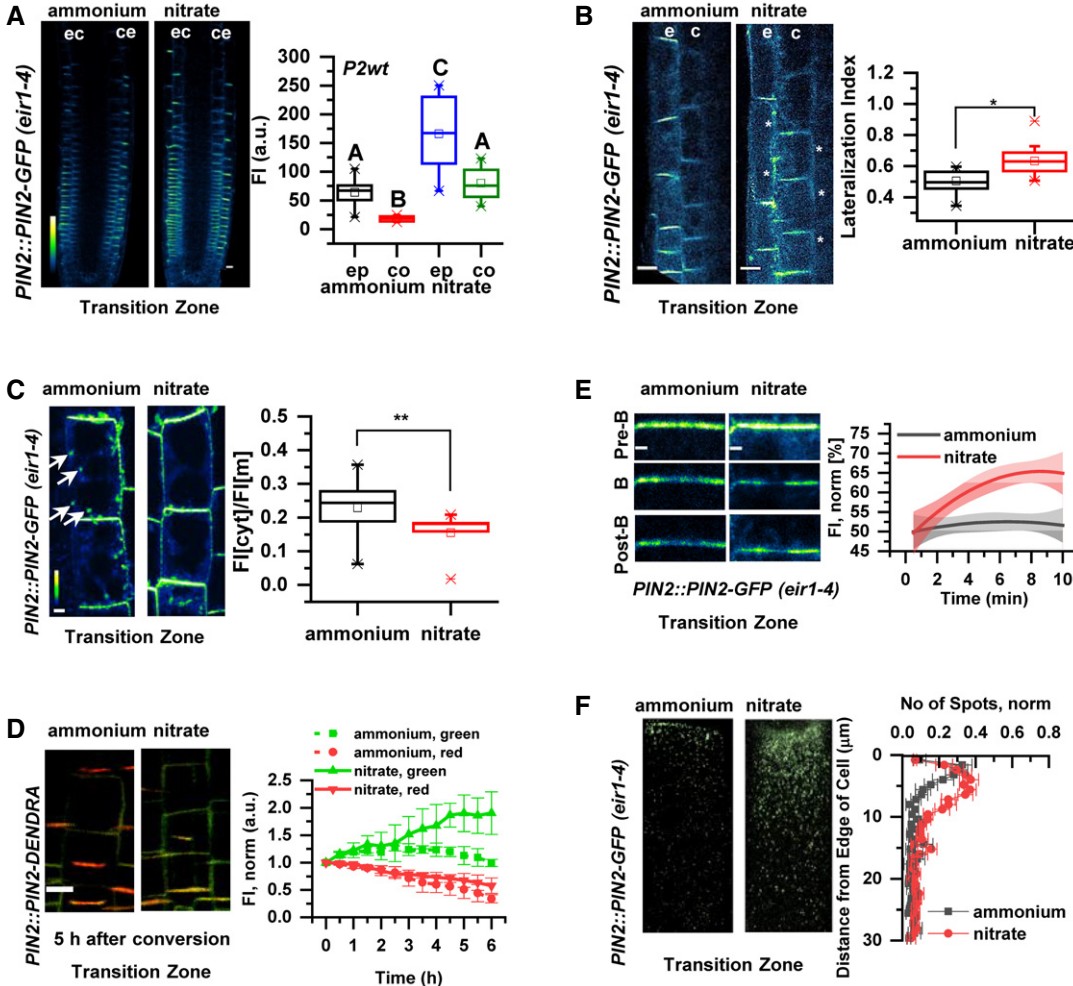

**Figure 4.  PIN2 protein abundance, polarity, and vacuolar trafficking in roots on ammonium or nitrate-containing medium.**

A   Pseudo-colored, optical longitudinal cross sections of 5 DAG roots expressing *PIN2::PIN2-GFP*, *eir1-4* 12 HAT to ammonium or nitrate supplemented media. "e" denotes epidermis and "c" cortex, respectively. Color code represents GFP intensity from low (blue) to high (white) values. Scale bar = 20 μm. Box plots display the distribution of the cell membrane-derived PIN2-GFP fluorescence intensity (FI) values (in arbitrary units, a.u.) on ammonium (gray, epidermis (ep) and red, cortex (co), *n* = 15) and nitrate (blue, epidermis (ep) and green, cortex (co), *n* = 10) grown roots. 5 cells per root were analyzed. The statistical significance was evaluated with ANOVA at *P* < 0.05. The box chart components are defined as, box (25–75%), central band (median line), and central box (mean), and the range is within 1.5IQR.

B   Higher magnification of pseudo-colored confocal images of 5 DAG old roots expressing PIN2-GFP 12 HAT to ammonium or nitrate supplemented media. "e" denotes epidermis and "c" cortex, respectively. Color code represents GFP intensity from low (blue) to high (white) values. Scale bar = 12 μm. White stars mark PIN2-GFP protein localization on the lateral membranes. Box plots display lateralization index (fluorescent signal detected on apical/basal membranes divided by the signal value at inner/outer membranes) of roots on ammonium (*n* = 31 cells from 6 roots) or nitrate (*n* = 24 cells from 6 roots) supplemented medium. The statistical significance was evaluated with ANOVA at *P* < 0.05. The box chart components are defined as, box (25–75%), central band (median line), and central box (mean), and the range is within 1.5IQR.

C   Pseudo-colored PIN2-GFP signal in epidermal cells of 5 DAG old roots 12 HAT to ammonium or nitrate-containing media. White arrows point to PIN2-GFP containing intracellular vesicles. Box plots represent the ratio in fluorescent signal detected inside the cell vs. on the membranes (FI[cyt]/FI[m]). *n* = 6 roots per condition, 5 cells per root were analyzed. Scale bar = 5μm. The statistical significance was evaluated with ANOVA at *P* < 0.05. The box chart components are defined as, box (25–75%), central band (median line), and central box (mean), and the range is within 1.5IQR.

D   Microscopic images showing PIN2-Dendra fluorescent signal 5 h after photoconversion of PIN2-Dendra into its red form. Depletion of the red signal and recovery of the green signal over a 6 h period was followed in parallel in 5 DAG old roots 12 HAT to ammonium or nitrate supplemented media. Note the increase in the intensity of the green signal in roots transferred to nitrate. Graph represents the mean signal ± SD (*n* = 6 roots per condition, 20 cells per root analyzed). The experiment was repeated 3 times. Scale bar = 20 μm.

E   FRAP analysis of PIN2 protein mobility in *PIN2::PIN2-GFP* expressing epidermal cells 12 HAT to ammonium or nitrate. The graph shows polynomial regression fit with 95% confidence band of the mean signal recovery in the bleached region of interest (ROI) after background subtraction and normalization to photobleaching. Data are derived from 3 biological replicates, each consisting of 5 membranes from 3 different roots. Scale bar = 2 μm.

F   Representative 3D SIM microscopic images of 10 DAG old epidermal cells expressing *PIN2-GFP* 12 HAT to ammonium or nitrate-containing media. Green dots represent PIN2-GFP on the lateral cell surface (polar domain) of epidermal cells in the transition zone. Graph represents the number of GFP-positive spots along a 30 μm long region starting at the apical side of the cell (8 cells per 4 roots and 9 cells per 4 roots) were analyzed per treatment, experiment was done 3 times. Error bars represent mean + SE. Note the effect of ammonium vs. nitrate on the distribution of the PIN2-GFP spots.

Source data are available online for this figure.

mechanism involved in adjustment of the PIN2 pattern in response to the available N source. In epidermal cells on $NH_4^+$ when compared to $NO_3^-$ supplied roots, the ratio between intracellular vs. PM-located PIN2-GFP was shifted in favor of intracellular localization and frequently endosomal vesicles with PIN2-GFP signal could be detected (Fig 4C, Movie EV3). This indicates that dynamics of PIN2 subcellular trafficking might be altered on the basis of the N source. To assess whether in $NH_4^+$ vs. $NO_3^-$ supplied roots, accumulation of PIN2 at the PM is the result of a changed balance between endo- and exocytosis, we analyzed *pPIN2::PIN2-Dendra* seedlings. The irreversible photoconversion of the Dendra fluorochrome by UV light from its green form to red allowed us to follow the impact of the N source on the subcellular fate of PIN2. By monitoring the PIN2-Dendra signal after photoconversion (red signal) vs. the newly synthesized PIN2-Dendra (green signal) in real time, we could evaluate the kinetics of PIN2 internalization from the PM and delivery of the *de novo* synthesized PIN2-Dendra proteins. We found that the kinetics of the photo-converted PIN2-Dendra (red signal) at the PM in either $NH_4^+$ or $NO_3^-$ were not statistically different, indicating that the internalization of PIN2 is not affected by the N source. Nevertheless, recovery of the newly synthesized PIN2-Dendra (green signal) was significantly enhanced in $NO_3^-$ when compared to $NH_4^+$ supplied roots (Figs 4D and EV3C). Considering that different sources of N did not have significant impact on *PIN2* transcription (Fig EV3A), these results suggest that recycling or secretion of PIN2 to the PM is more promoted in $NO_3^-$ supplied roots than in those on $NH_4^+$. To further examine the impact of N source on the delivery of PIN2 to the PM, we performed Fluorescence Recovery After Photobleaching (FRAP) analyses on the apical membrane of the cell. Lateral diffusion of PIN proteins at the PM is negligible (Łangowski *et al*, 2016), thus PIN2-GFP signal recovery after photobleaching can be correlated with the delivery of PIN2 protein to the PM. In epidermal cells of $NO_3^-$ supplied roots, PIN2-GFP signal recovered significantly faster as compared to roots supplied with $NH_4^+$ (Fig 4E), thus strongly suggesting that delivery of PIN2 toward the PM is differentially regulated by specific forms of N source.

Finally, to examine whether the above described different recycling behavior of PIN2 has an impact on the establishment of its apical polar domain, we performed super-resolution imaging employing three-dimensional structured illumination microscopy (3D-SIM). In roots supplemented with either $NH_4^+$ or $NO_3^-$, PIN2-GFP accumulated at the apical edge of epidermal cells to the same level. However, in $NH_4^+$ supplemented roots, number of the PIN2-GFP-positive particles decreased with distance from the cell edge significantly more than in roots supplied with $NO_3^-$ (Fig 4F).

In summary, these results suggest that PIN2 subcellular trafficking, and in particular, the delivery of PIN2 to the PM is differentially adjusted according to the N source. In $NO_3^-$ supplied roots, trafficking of PIN2 to the PM and the lateral domains is more promoted when compared to $NH_4^+$ supplemented roots.

## Nitrogen-dependent PIN2 phosphorylation fine-tunes intracellular dynamics and membrane polarity of PIN2

Post-translational modifications including phosphorylation are regulatory cues with significant impact on the intracellular trafficking and polar membrane localization of PIN proteins (Barbosa *et al*, 2018). Phosphoproteome analysis of samples with either $NH_4^+$ or

$NO_3^-$ as their N source revealed that PIN2 was among the proteins exhibiting an altered pattern of phosphorylation in response to $NH_4^+$ (Vega *et al*, 2020). Ser439 located at the very end of the PIN2 cytoplasmic loop (C-loop) was identified as a potential target for differential phosphorylation, where a reduction of phosphorylation in $NO_3^-$ conditions was detected compared root supplied with either $NH_4^+$ or KCl (Vega *et al*, 2020). Multiple sequence alignment revealed that this Ser439 residue is highly specific to PIN2 (Fig EV3E). Interestingly, amino acid sequence alignment of PIN2 orthologues indicated that Ser439 is highly conserved in the PIN2 or PIN2-like clade across plant species including gymnosperms, mono-, and dicotyledonous plants (Fig EV3F).

To examine a role of this specific, uncharacterized phosphosite in subcellular dynamics and function of PIN2, we introduced amino acid substitutions S439D and S439A to achieve either gain or loss of phosphorylation variants of PIN2, respectively. *PIN2::PIN2$^{S439D}$-GFP* and *PIN2::PIN2$^{S439A}$-GFP* constructs were introgressed into the *eir1-4* mutant line. The phospho-variant version PIN2$^{S439D}$-GFP, like PIN2-GFP, accumulated at the PM of epidermal and cortex cells significantly more in roots supplied with $NO_3^-$ than $NH_4^+$ (Fig 5A, B and D). Interestingly, the amount of the PM localized phospho-dead PIN2$^{S439A}$-GFP in $NH_4^+$ -supplied roots was significantly higher when compared to PIN2-GFP and PIN2$^{S439D}$-GFP, and only a slight increase in epidermal cells could be detected in response to $NO_3^-$ supply (Fig 5A, C and D). Furthermore, in cortex cells at the TZ, reduced lateralization of PIN2$^{S439D}$-GFP on $NO_3^-$ supplemented medium could be observed. PIN2$^{S439A}$-GFP lateralized toward outer and inner PMs irrespective of the N source thus phenocopying the PIN2-GFP pattern in $NO_3^-$ supplied roots (Fig 5E–G). Altogether, these results suggest that the phosphorylation status of PIN2 on S439 direct PIN2 trafficking toward the PM and PIN2 lateralization under varying N sources. Enhanced accumulation of the phospho-dead PIN2 variant at the PM and the lateral domains in roots supplied with $NH_4^+$ mimicked the behavior of the wild-type PIN2 in $NO_3^-$ supplied roots, supporting the role of nitrate mediated dephosphorylation as an important mechanism underlying adjustment of PIN2-mediated auxin transport to the source of N.

## Nitrogen-dependent PIN2 phosphorylation regulates PIN2-mediated root growth

Next, we examined the impact of PIN2 phosphorylation status on the root growth adaptations to different N source. To evaluate functionality of PIN2-GFP constructs with phosphosite substitutions, we analyzed their ability to rescue the agravitropic phenotype of *eir1-4*. *PIN2::PIN2$^{S439D}$-GFP* as well as *PIN2::PIN2$^{S439A}$-GFP* constructs were able to rescue the agravitropic phenotype of the *eir1-4* mutant (Appendix Fig S6A), indicating that the overall activity was maintained in both mutated variants. However, measurements of roots 6, 24, and 96 HAT on either $NH_4^+$ or $NO_3^-$ supplemented media revealed that modulation of PIN2 phosphorylation status interferes with the flexible adjustment of root growth to N source (Movies EV4 and EV5). Roots of *PIN2::PIN2$^{S439D}$-GFP, eir1-4* exhibited enhanced growth already 6 h after transfer on $NO_3^-$ when compared to $NH_4^+$ supplemented medium (Fig EV4A); however, when compared to control seedlings, the enhancement of root growth by $NO_3^-$ was less pronounced 4 DAT (Appendix Fig S6B). This suggests that PIN2$^{S439D}$-GFP is partially able to mediate distinct root growth

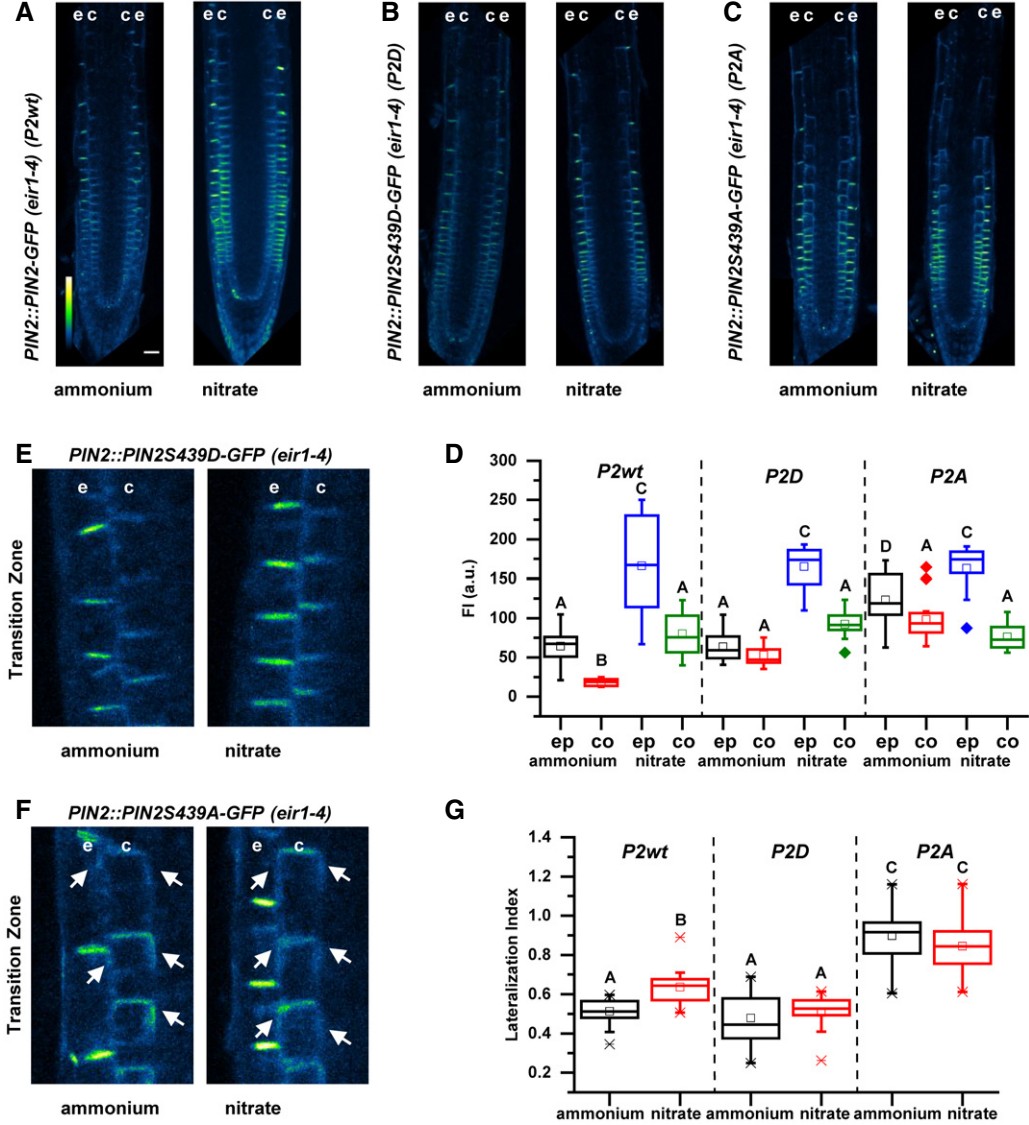

**Figure 5. Impact of Ser439 on PIN2 localization in roots supplemented with ammonium or nitrate.**

A–C  Pseudo-colored, optical longitudinal cross sections of 5 DAG roots expressing (A) PIN2-GFP (*PIN2::PIN2-GFP, P2wt*) (B) *PIN2S439D-GFP* (*PIN2::PIN2S439D-GFP, P2D*) and (C) *PIN2S439A* (*PIN2::PIN2S439A-GFP, P2A*)—all in *eir1-4* background – 12 HAT to ammonium or nitrate supplemented media. "e" denotes epidermis and "c" cortex, respectively. Color code represents GFP intensity from low (blue) to high (white) values. Scale bar = 50 μm.

D  Box plots display the distribution of the cell membrane-derived PIN2-GFP fluorescence intensity (FI) values (in arbitrary units, a.u.) in roots transferred to ammonium ((gray, epidermis (ep) and red, cortex (co)) and to nitrate (blue, epidermis (ep) and green, cortex (co)). 5 cells per roots were analyzed in at least 9 roots per genotype per treatment. The statistical significance was evaluated with ANOVA at $P < 0.05$. The box chart components are defined as, box (25–75%), central band (median line), and central box (mean), and the range is within 1.5IQR.

E, F  Microscopic images of 5 DAG old roots expressing (E) *PIN2::PIN2S439D-GFP* and (F) *PIN2::PIN2S439A-GFP* 12 HAT to ammonium or nitrate amended media. "e" denotes epidermis and "c" cortex, respectively. White arrows point to PIN2-GFP protein localization on the lateral membranes.

G  Box plots display lateralization index (fluorescent signal detected on apical/basal membranes vs. inner/outer membranes) of *P2wt, P2D*, and *P2A* roots transferred to ammonium (gray) or nitrate (red) supplemented medium. At least 24 cells from 5 roots were analyzed per genotype per treatment. The statistical significance was evaluated with ANOVA at $P < 0.05$. The box chart components are defined as, box (25–75%), central band (median line), and central box (mean), and the range is within 1.5IQR.

Source data are available online for this figure.

responses to different N sources. Roots of *eir1-4* expressing *PIN2:: PIN2S439A-GFP* exhibited delay in adjusting growth to $NO_3^-$ provision and no significant increase in length 6 and 24 HAT to $NO_3^-$ when compared to $NH_4^+$ could be detected (Fig EV4A; Appendix Fig S6B).

Intriguingly, although *PIN2::PIN2S439D-GFP* partially recovered the ability of *eir1-4* roots to adjust elongation growth to N source, in depth analysis of epidermal and cortex cell files revealed intriguing differences when compared to control roots. In *PIN2::PIN2S439D-GFP*;

*eir1-4* roots—irrespective of the N source—length of epidermal cells steeply increased with distance from QC, whereas cortex cells underwent slow steady elongation. Similar asynchronous growth patterns in epidermal and cortex cell files were observed in $NH_4^+$, but not in $NO_3^-$ supplied *PIN2::PIN2-GFP; eir1-4* and Col-0 roots, indicating that *PIN2::PIN2^{S439D}-GFP* is not able to recover all aspects of root adaptation to varying N supply (Fig EV4B and C compared to Appendix Fig S6C and Fig 1C). In *PIN2::PIN2^{S439A}-GFP* roots irrespective of the N source, shallow slope of epidermal cell length was detected, which resulted in synchronized growth patterns of epidermal and cortex cell files, resembling those observed in *PIN2::PIN2-GFP, eir1-4* and Col-0 roots supplied with $NO_3^-$ (Fig EV4D and E compared to Appendix Fig S6C and Fig 1D). Thus, *PIN2::PIN2^{S439A}-GFP,0 eir1-4* roots supplemented with $NH_4^+$ acquired features typical for Col-0 roots supplied with $NO_3^-$.

In summary, the PIN2^{S439D}-GFP phospho-variant partially recovers enhanced elongation growth of *eir1-4* roots, but it is unable to synchronize the patterns of epidermis and cortex elongation in response to $NO_3^-$. Unlike PIN2^{S439D}-GFP, PIN2^{S439A}-GFP is unable to rescue sensitivity of *eir1-4* roots to $NO_3^-$ stimulatory effect on root elongation growth and to synchronize the patterns of epidermis and cortex elongation growth, irrespective of N source. Taking together, these results indicate that N-dependent regulation of phosphorylation status of PIN2 at S439 is a part of a complex mechanism underlying root growth adaptation to specific N source, which involves coordination of the balance between cell proliferation and elongation in the adjacent epidermis and cortex tissues. Reduced phosphorylation of PIN2 at S439 promoted by $NO_3^-$ leads to alterations in subcellular localization of the auxin efflux carrier and subsequent adjustment of the root growth pattern to the nitrogen source.

### NRT1.1-mediated nitrate transport and sensing is involved in the root growth adaptation to nitrogen source

Our results suggest that the adaptation of root growth to the differing nitrogen sources involves tissue-specific fine-tuning of auxin fluxes achieved by nitrogen source-dependent phosphorylation of PIN2. To verify that growth adaptation of roots to either $NH_4^+$ or $NO_3^-$ represents a specific, nitrogen source-dependent response, and they are not the consequence of a more general stress condition caused by the experimental setup, we performed three sets of experiments (Fig EV5A–D). First, we investigated root growth after transfer from $NH_4^+$ to $NO_3^-$ in a set of nitrate transceptor mutants. NRT1.1 plays a fundamental role in both $NO_3^-$ acquisition and signaling (Remans *et al*, 2006; Krouk *et al*, 2010; Mounier *et al*, 2014). It is a dual-affinity transporter, which can facilitate $NO_3^-$ assimilation over a wide range of $NO_3^-$ concentrations and it also acts as a $NO_3^-$ sensor, which regulates the gene expression of other $NO_3^-$ transporters such as NRT2 (Sun & Zheng, 2015). To explore the impact of the NRT1.1-mediated transport and sensing in root growth responses to specific sources of nitrogen, we examined different mutant alleles of NRT1.1. These were as follows: *nrt1.1/chl1-5* deletion mutant (defective in both low- and high-affinity $NO_3^-$ transport) (Tsay *et al*, 1993); *chl1-9* mutant (defective in $NO_3^-$ uptake, but displays a normal primary $NO_3^-$ response) (Ho *et al*, 2009); T101A loss of phosphorylation allele (which displays reduced high-affinity $NO_3^-$ uptake); and T101D phosphomimetic-mutant allele (defective in the low-affinity $NO_3^-$ uptake) (Ho *et al*,

2009). We found that root growth of wild-type roots was significantly enhanced 12 HAT from $NH_4^+$ compared to $NO_3^-$ containing medium (Fig EV5A). Unlike wild type, roots of the *nrt1.1* mutants were unable to enhance growth in response to $NO_3^-$ (Fig EV5A). These observations suggest that a fully functional NRT1.1, with both intact transport and sensing functions mediate the acceleration of primary root growth by $NO_3^-$ availability.

To further explore whether this NRT1.1-mediated root growth adaptation in response to nitrogen source might involve modulation of PIN2 accumulation at the plasma membrane, we analyzed PIN2 protein abundance in the *chl1-5* background. Quantification of PIN2-specific immunostaining revealed no increase in PIN2 membrane abundance in the *chl1-5* background compared to the *PIN2::PIN2-GFP/eir1-4* line in roots transferred to $NO_3^-$ (Fig EV5B). Altogether, these findings suggest that both PIN2 accumulation at the PM and the enhanced root growth in response to $NO_3^-$ are mediated through the NRT1.1 transceptor, underlining the nitrogen source-dependent aspects of our observations.

Finally, we tested whether alteration of protein abundance of the PIN2 protein in roots supplemented with $NH_4^+$ when compared to $NO_3^-$ represents a PIN2 specific response. Hence, we analyzed the expression of two other membrane proteins PIN1 from the PIN family (Gälweiler *et al*, 1998) and PIP2 [(Plasma membrane intrinsic protein 2-1; (Johanson *et al*, 2001)]. Seedlings were grown on $NH_4^+$ for 5 DAG and transferred onto media containing either $NH_4^+$ or $NO_3^-$. Fluorescence intensity measurements of PIN1-GFP and PIP2-GFP signals at the PM revealed that unlike PIN2, none of the analyzed proteins showed reduced abundance on $NH_4^+$ compared to $NO_3^-$ (Fig EV5C and D), indicating that modulation of PIN2 abundance at the PM is a specific and adaptive response to the nitrogen source.

### An experimentally derived quantitative model predicts nitrogen-dependent coordination of root growth

Experimental findings suggest that nitrogen-dependent fine-tuning of polar auxin transport through the regulation of PIN2 phosphorylation status could coordinate the growth of adjacent tissues and thereby steer the root growth. To mechanistically understand nutrient effect on plant root growth, we developed a multilevel computer model of epidermis and cortex tissues. The complete scheme of the model components can be found in Appendix Fig S7, Fig 6A and a full description of the model is provided in the Materials and Methods section. The model integrates the experimental observations of N source-dependent effects on PIN2 accumulation at the PM and previously shown auxin-dependent degradation of PIN2 (Abas *et al*, 2006; Baster *et al*, 2013). As a source of auxin, we tested two likely scenarios (i) a uniform source of auxin along the epidermis (Model A) or (ii) flow of auxin from lateral root cap (LRC) and QC into epidermis (Petersson *et al*, 2009; Tian *et al*, 2013) (Model B). In addition, other potential scenarios were tested by introducing independent auxin inputs in epidermis (Models C and D), and however, these models showed a poor performance compared to model B when faced with experimental measurements and thus were less favorable (see Materials and Methods and Appendix Fig S7F).

In our model, PIN2 polarity and auxin distribution as well as cell length and number of cell division resulted purely from predictions of the model. To test the model, we compared experimental

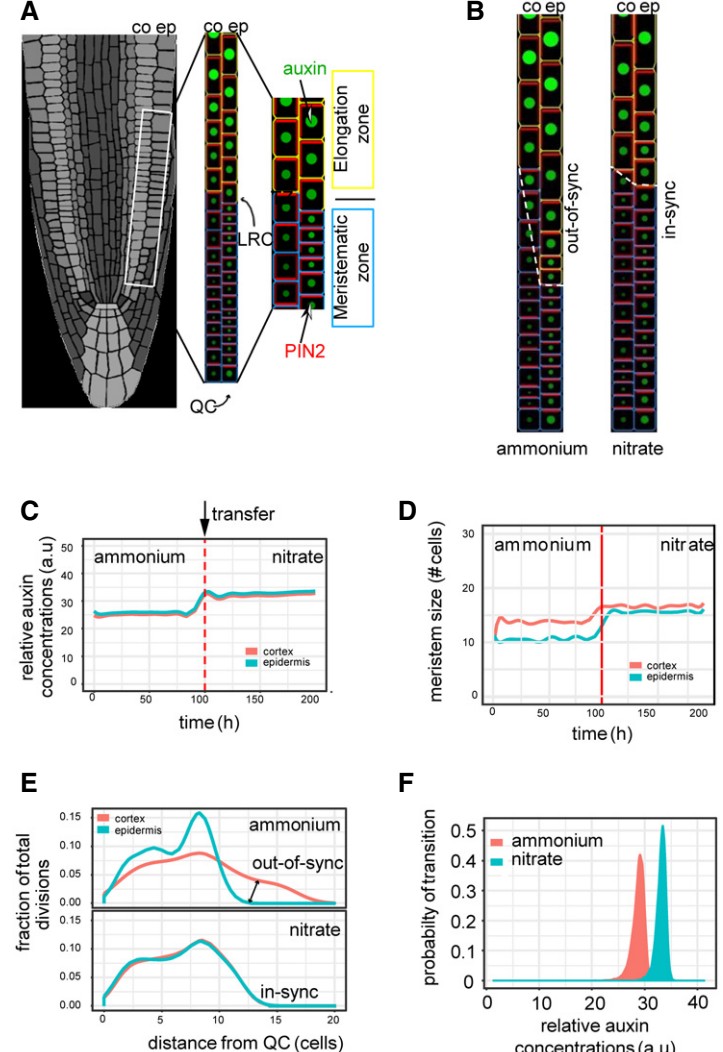

**Figure 6.  Dynamic computer model of root growth predicts nitrogen source-dependent effect on cell growth dynamics, auxin distribution, and root zonation.**

A      Schematics of the root model with epidermis (ep) and cortex (co) tissues. Meristematic and elongating cells are shown with blue and yellow walls, respectively. Auxin levels are represented by green circle size and red bars reflect the PIN2 amounts. Auxin is supplied from Lateral Root Cap (LRC) and QC (Model B).

B      Steady-state snapshots from model simulation with ammonium (left panel) and with nitrate (right panel). Note out-of-sync growth patterns (dashed white line) in ammonium.

C, D  Model simulation representing the effect of the transition from ammonium to nitrate (denoted by a red dashed line) on the relative level of auxin (C) and meristem size measured as distance from QC (D).

E      Model predictions display the fraction of total cell division events per cell in the meristem along in the two N source. Note cell division is out-of-sync in ammonium, producing altered growth of the root.

F      Experimentally derived relative auxin level threshold triggering cell elongation depends on the actual N content of the root.

observations of PIN2 distributions (Movie EV6), cell length measurements (Fig 1), and auxin content (Fig 2) with those predicted by computer model simulations. The initial Model A failed to recapitulate experimental data (Appendix Fig S7B–D), indicating that the auxin source assumption may not be correct and/or there are missing components, which were not considered in this model. Model B, which unlike Model A, integrates flow of auxin from the QC and the LRC into the epidermis, and in addition, it implements correlation between cell distance from the QC with both increased PIN2 trafficking and PIN2 degradation (Appendix Fig S7E), was able

to recapitulate PIN2 and auxin distributions as well as cell length across the meristem (Appendix Fig S7A–F). To comprehend a necessity for these two essential components in our model, we closely inspected the relation between auxin activity levels and PIN2 fluorescence in our experimental dataset in roots supplemented with $NO_3^-$ or $NH_4^+$. Our analysis revealed that for the same auxin activity two different PIN2 levels were observed in both the cortex and the epidermis that was dependent on distance from the QC—a component missing in Model A (Appendix Fig S7A). This eminently bistable feature was important to guarantee the synchrony of cell

elongation between adjacent tissues as this feature was compromised in $NH_4^+$ grown roots that showed an asynchronous elongation of adjacent cortex and epidermal cells (Appendix Fig S7A). Notably, Model B could successfully capture this relation (Appendix Fig S7C and D). Finally, we coupled auxin activity to cell division and elongation and simulated our root model in both $NH_4^+$ and $NO_3^-$ regimes (Fig 6A and B). As for previous simulations, the computer model of root growth does not include neither fixed auxin levels nor pre-patterned PIN2 polarization and was capable of recapitulating *in planta* root growth patterns in those different N sources (Fig 6B, Movies EV7 and EV8). Furthermore, model predictions such as lateral auxin distribution (Fig 6C), meristem length (Fig 6D), and proliferation dynamics (Fig 6E) are in a fair agreement with experimental results (Figs 1 and 2). Importantly, our model (Appendix Fig S8A–C) predicts mechanistic principles of the growth in synchrony such as coordinated cell divisions in both epidermis and cortex tissues further away from QC (Fig 6E) and lateral auxin transport through PIN2 between cortex and epidermis near the transition zone (Fig 6C).

Mechanisms that trigger the transition from meristematic activity to cell elongation are not well understood (Velasquez *et al*, 2016). Auxin plays a fundamental role in the establishment of the TZ (Chapman & Estelle, 2009; Kong *et al*, 2018). Our model could predict a precise threshold of auxin levels that was necessary to determine the transition to elongation. This auxin threshold is dynamic as it depends on the actual N source; in particular, higher levels of auxin were required to advance cell elongation on $NO_3^-$ (Appendix Fig S7F). Taken together, we have developed a quantitative experimentally supported computer model of root growth in different N sources that was capable of recapitulating all experimental observations as well as generating new predictions that could broaden our understanding of root growth mechanisms in the dynamic environment.

# Discussion

Ammonium and nitrate represent major inorganic forms of N absorbed by plants. Since the distribution of these N sources in the soil is very heterogeneous(Jackson & Caldwell, 1993), plants tend to maximize the N exploitation by flexible modulation of root system architecture (Nacry *et al*, 2013). Although distinct impacts of $NH_4^+$ and $NO_3^-$ on the root system growth and development have been already demonstrated (Jia & Wirén, 2020), molecular mechanisms how spatio-temporal changes in N resource impact on root growth are scarcely reported.

Root growth is determined by the production of new cells at the apical meristem and their rapid elongation need to be well-coordinated across diverse cell types of the root organ. Conversion from proliferative to elongation phases occurs as cells pass through the TZ where they undergo complex cyto-architectural re-arrangement (Kong *et al*, 2018). Hence, alterations of root growth kinetics might result from modulation of any of these growth-determining processes. We show that replacement of $NH_4^+$ for $NO_3^-$ has a rapid impact on root growth kinetics and in particular progression of cells through individual root zones. While in roots supplied with $NH_4^+$, proliferative capacity of epidermal cells is attenuated in closer distance to the QC, which led to their earlier and rapid transition to

elongation phase when compared to cortex, provision of $NO_3^-$ promotes proliferation and steady elongation of epidermal cells, which results in well-synchronized growth patterns of epidermis and cortex. Hence, adaptation of primary roots to different sources of N encompasses a tissue-specific modulation of cell proliferation and cell growth kinetics.

Auxin is an essential patterning cue during plant growth and development. A number of recent studies have demonstrated that levels and distribution of this hormone have instructive function in many aspects of root growth including the root apical meristem patterning, its size determination, transition of meristematic cells into the elongation phase and capacity of cells to elongate (Velasquez *et al*, 2016; Di Mambro *et al*, 2017; Barbez *et al*, 2017). Whereas the exit of cells from meristematic zone was associated with local auxin minima that has been proposed to define the transition zone (Di Mambro *et al*, 2017), increase of auxin signaling along the longitudinal root growth axis correlated with cell wall acidification as a potential driving force of cell elongation (Barbez *et al*, 2017).

Experimental measurements supported by a quantitative computational model indicate that adjustment of root growth dynamics in different N regimes is dependent on the precise modulation of auxin transport routes between cortex and epidermis. The steep increase of auxin activity correlating with earlier attenuation of proliferation activity in the epidermis and transition of cells into the elongation phase was eminent in roots grown on $NH_4^+$. In contrast, shallow slopes of the auxin activity in both epidermis and cortex corresponded with delayed, gradual transition of epidermal cells into elongation phase in roots supplemented with $NO_3^-$, showing a tight growth synchronization with adjacent cortex tissues. Based on these observations, we demonstrate that a flexible modulation of auxin activity in response to varying sources of N is largely consistent with described impact of auxin on key events defining root growth such as transition into elongation growth and kinetics of elongation.

Delivery of auxin in outer tissues including the cortex and the epidermis is largely mediated by the PIN2 auxin efflux carrier (Luschnig *et al*, 1998; Müller *et al*, 1998). While PIN2 dependent basipetal transport of auxin is instructive for elongation growth and root gravity bending (Su *et al*, 2017), PIN2-mediated reflux to inner tissues has been associated with maintenance of root meristem size (Blilou *et al*, 2005). Measurements of the auxin transport revealed that replacement of $NH_4^+$ by $NO_3^-$ significantly enhances flow of auxin in the basipetal direction which correlates with increased PIN2 activity near the transition zone. Loss of PIN2 activity not only interferes with the $NO_3^-$ stimulated transport of auxin toward the shoot, but also severely affects adaptive responses of roots to this N source. Furthermore, model predictions based on these experimental measurements suggest a bi-stable relationship between auxin levels and PIN2 activity and cell elongation that is enhanced in $NO_3^-$, which could explain why roots grown on nitrate can coordinate their growth by passing auxin between cortex and epidermal cells in a synchronous manner. Furthermore, our model confirmed the necessity for self-emerging communication between cortex and epidermis via auxin with quantitative computer simulations of root growth under different N conditions.

Dynamic, N source-dependent accumulation and polarization of PIN2 at the PM, but unchanged *PIN2* transcription, pointed at post-transcriptional regulatory mechanism underlying adaptation of

basipetal auxin transport to N supply. Replacement of $NH_4^+$ by $NO_3^-$ promoted accumulation of PIN2 at the apical PM of epidermal and cortex cells as well as to the lateral sides.

Phosphorylation has been recognized as a prominent post-translational modification of PIN proteins that determines their polar membrane localization and activity (Zwiewka *et al*, 2019). Unexpectedly, genome-wide analysis of the phosphoproteome during early phases of root adaptation to provision of $NO_3^-$ (Vega *et al*, 2020) retrieved PIN2 among differentially phosphorylated proteins. Serine 349 of PIN2 in *Arabidopsis* found to undergo a rapid dephosphorylation after replacement of $NH_4^+$ by $NO_3^-$. The PIN2S439 phosphosite was not completely unknown: It was originally identified as differentially phosphorylated during lateral root morphogenesis (Zhang *et al*, 2013). It is positioned in the hydrophilic loop domain of the PIN2 protein and is an evolutionarily conserved residue in the PIN2 or PIN2-like clade across species including gymnosperms, mono-, and dicotyledonous plants, suggesting that PIN2 might be universally involved in other plant species adaption strategies to the changing N sources by means of its post-translational (phosphorylation) mechanism. The analyses of PIN2 phospho-variants suggest that N source-dependent regulation of PIN2 phosphorylation status has a direct impact on the flexible adjustment of PIN2 membrane localization and polarity and thereby adaptation of root growth to varying forms of N supply. Notably, inability of the mutants defective in the NRT1.1-mediated transport and sensing of $NO_3^-$ to enhance their root growth and accumulate PIN2 at the PM after transfer to $NO_3^-$ supports a role of the nitrate regulatory components in these adaptive responses.

In summary, our work provides molecular insights into how signals of nutrient availability are translated into root growth and developmental adaptive responses.

# Materials and Methods

### Plant material

*Arabidopsis thaliana* (L.) Heynh plants were used in this work. The transgenic lines *W131Y* (Geldner *et al*, 2009), *PIN2::PIN2-GFP* (Vieten *et al*, 2005) in *eir1-4* background, *PIN2::PIN2S439D-GFP*, *PIN2::PIN2S439A-GFP* (Vega *et al*, 2020) were introduced into *eir1-4* background; *PIN2::PIN2-Dendra* (Jásik *et al*, 2013), *R2D2* (Brunoud *et al*, 2012), *DII-VENUS* (Brunoud *et al*, 2012), *mDII-VENUS* (Brunoud *et al*, 2012), *PIN2::nls-GFP* (Salanenka *et al*, 2018), *DR5::LUC* (Ulmasov *et al*, 1997), *DR5::RFP* (Marin *et al*, 2010), *CyclinB1::GUS* (Colón-Carmona *et al*, 1999), *PIN1::PIN1-GFP* (Gälweiler *et al*, 1998), *PIP2::PIP2-GFP* (Johanson *et al*, 2001), *CHL1T101A/D* in *chl1-5* background (Ho *et al*, 2009) and the T-DNA mutant lines *eir1-4*, *chl1-5* (Tsay *et al*, 1993), and *chl1-9* (Ho *et al*, 2009) were described previously. *DII-VENUS* and *mDII-VENUS* in *eir1-4* background lines were obtained by manual hand pollination of the individual lines.

### Growth conditions

Seeds of *A. thaliana* were surface-sterilized by 70% ethanol and sown on a modified Murashige and Skoog (MS) medium—boric acid 6.2 mg/l, calcium chloride (anhydrous) 332.2 mg/l, cobalt chloride $(6H_2O)$ 0.025 mg/l, cupric sulfate $(5H_2O)$ 0.025 mg/l, $Na_2EDTA$

$(2H_2O)$ 37.26 mg/l, ferrous sulfate $(7H_2O)$ 27.8 mg/l, magnesium sulfate (anhydrous) 180.7 mg/l, molybdic acid (disodium salt $2H_2O$) 0.25 mg/l, potassium iodide 0.83 mg/l, potassium phosphate (monobasic, anhydrous) 170 mg/l, zinc sulfate $(7H_2O)$ 8.6 mg/l—which contained 0.5 mM Ammonium Succinate (Santa Cruz Biotechnology) (76 mg/l) as a nitrogen source and supplemented with 0.1% sucrose and 1% agar (Type E, Sigma A4675), pH = 5.8. The nitrate amended media contained 5 mM potassium nitrate (505 mg/l) instead of 0.5 mM ammonium succinate. Seeds were stratified at least for 3 days and grown for 4–14 days at 21°C in a 16 h light/8 h dark cycle.

### Root growth and root length analysis

Seven-day-old light-grown seedlings were transferred to either ammonium or nitrate amended plates and scanned on a daily basis for 7 days on an Epson Perfection V700 flatbed scanner. Root growth (root length changes over a given period of time) and root length were measured manually using Fiji (v1.52).

### Cell elongation and cell length analysis

Cell elongation was measured after 12 h exposure to either to ammonium or nitrate manually with the software Fiji (v1.52).

For cell length analysis, confocal microscopic images of propidium iodide-stained *PIN2::PIN2-GFP*, *PIN2::PIN2S439A-GFP*, *PIN2::PIN2S439D-GFP*, Col-0, and *eir1-4* roots were used, and the length of each cell in different cell files (epidermis and cortex) was measured manually using Fiji (v1.52).

### Imaging and image analysis

5 DAG seedlings were mounted on a slice of MS medium—containing either 0.5 mM ammonium or 5mM nitrate—placed into a chambered coverslip (Lab-Tek) and imaged with Zeiss LSM700, LSM800, or LSM880 inverted confocal microscopes equipped either with a 20×/0.8 Plan-Apochromat M27 objective or a 40× Plan-Apochromat water immersion objective. Fluorescence signals for GFP (excitation 488 nm, emission 507 nm), YFP (excitation 514 nm, emission 527 nm), PI (excitation 536 nm, emission 617 nm), and DAPI (excitation 405 nm, emission 461 nm) were detected. A LaVision 2-Photon Inverted TriM Scope II from LaVision Biotec with a FLIM X16 TCSPC detector from LaVision Biotec equipped with a Olympus UApo N340 40xW, NA 1.15 was also used. Roots were observed 12 h after transfer to ammonium or nitrate supplemented media. Long time-lapse imaging was performed using a vertically oriented LSM700 microscope as described previously (von Wangenheim *et al*, 2017).

For image quantification (R2D2, DII-Venus, mDII-Venus, PIN2-GFP fluorescence intensity measurements), maximum intensity projections of confocal pictures were used. Images were handled and analyzed with Fiji (v1.52) and Adobe Photoshop (Adobe Creative Cloud).

### PIN2-DENDRA photoconversion and FRAP experiments

PIN2-DENDRA experiments were executed as previously described (Salanenka *et al*, 2018). Briefly, photoconversion of 5 DAG seedlings

expressing *PIN2-Dendra* into its red form induced by illuminating the region of interest with UV light and the depletion of the red and re-appearance of the green signals in ammonium or nitrate transferred *Arabidopsis* roots was followed over time using a vertically oriented LSM700 microscope. Observation of roots initiated 10–20 min after transfer and images was recorded every 20 min (9 stacks/root/recording). The experiment was repeated three times, and each experiment consisted of imaging six roots per condition. Image analysis was performed using Fiji (v1.52). Red and green fluorescent signal changes were measured on 10–10 individual cell membranes in the TZ over a period of 6 h. FRAP experiments were performed as described previously (Glanc *et al*, 2018). Briefly, individual membranes of 5 DAG old *PIN2-GFP* expressing *Arabidopsis* roots transferred either to ammonium or nitrate were bleached using the 488nm laser of a Zeiss LSM800 confocal microscope according to its built-in bleaching protocol. Recovery of the PIN2-GFP signal at the bleached areas was followed for 10 min, and quantification of fluorescence recovery was measured using Fiji (v1.52).

### PIN2 immunodetection and staining of nuclei

For PIN2 immunostaining, 5 DAG *Arabidopsis* roots were handled as previously described (Pasternak *et al*, 2015). Briefly, fixation was performed using 2% PFA (in 1×MTSB) supplemented with 0.1% Triton X-100, followed by hydrophilization using MeOH 100% (65°C, 10 min), cell wall digestion using 0.2% Driselase and 0.15% Macerozyme in 2 mM MES, pH 5.0 (37°C, 40 min), and membrane permeabilization using 3% NP-40, 10 % DMSO in 1× MTSB (37°C, 20 min). Anti-PIN2 (1:100) was used as a primary antibody (37°C, 120 min). Alexa Fluor 488 goat anti-rabbit IgG H + L (Thermo Fischer Scientific) was used as secondary antibody (1:800) (37°C, 60 min). Finally, samples were mounted in VECTASHIELD® Antifade Mounting Medium with DAPI (4′,6-Diamidino-2-Phenylindole, Dihydrochloride). Images were obtained using an LSM800 microscope.

### Quantification of R2D2, DII-VENUS, and mDII-VENUS fluorescence signal in *Arabidopsis* roots

*R2D2* combines *RPS5A*-driven *DII* (*DII* domain of the *INDOLE-3-ACETIC ACID28* (*IAA28*, *DII*) from *Arabidopsis*) fused to *n3×Venus* and *RPS5A*-driven *mDII* fused to *ntdTomato* on a single transgene (Brunoud *et al*, 2012; Liao *et al*, 2015). DII-VENUS is the domain II of IAA28 fused to the VENUS fast maturing YFP, and mDII-VENUS is the non-degradable form of DII-VENUS. The analysis of the fluorescence intensity of either R2D2, DII-VENUS, or mDII-VENUS expressing plants grown on ammonium containing and transferred on ammonium and nitrate-containing medium was performed on Maximum Intensity Projection of Z-stacks of root tips acquired with a Zeiss LSM 700 inverted laser-scanning microscope as described in (Di Mambro *et al*, 2017) with slight modifications.

To quantify the fluorescence signal in each cell per selected root tissue (epidermis and cortex) first, we positioned a segmented line over the nuclei in the corresponding tissues with the ROI manager tool of the software Fiji (v1.52) (Appendix Fig S2C). Next, we analyzed the fluorescence plot profiles of the different lines with the peak analyzer function of the software Origin (OriginLab Corporation) to find local maxima along the lines, which represented the

fluorescence value of the nuclei in the tissues. In case of R2D2, auxin distribution plots were derived by reciprocal mean values of the normalized n3xVenus/ntdTomato ratio. Relative auxin level data in each cell per tissue were graphed after data interpolation using the Origin built-in algorithm for smoothing.

### 3D SIM and polar density analysis of PIN2-GFP

Live *Arabidopsis* seedlings, which were incubated on either nitrate or ammonium amended medium for 6–8 h, were mounted on to coverslips as previously described by Johnson and Vert (Johnson & Vert, 2017) with the coverslips additionally fixed to the slide with nail polish. Cells in the elongation zone of the root epidermis were imaged using an OMX BLAZE v4 3D SIM (Applied Precision), as described (Hille *et al*, 2018). Briefly, a 60× 1.42 NA Oil Immersion objective and a 100 mW 488 laser was used to make optical sections in the Z dimension, in order to capture the totality of the lateral polar domain of the subject cell. Each Z-section image is based on 15 images generated from three different angles and five different SIM patterns and reconstructed using SOFTWORX software (Applied Precision).

A maximum projection of the Z-stack was used for analysis. Images were made binary and subjected to watershed segmentation using Fiji (Schindelin *et al*, 2012). PIN2 spots were then detected using TrackMate (Tinevez *et al*, 2017). The number of PIN2 spots was calculated in regions of interest (0.8 microns in width times the height of the cell) at distances sequentially further away from the polar end of the cell using a custom made Matlab script. The raw number of spots in each ROI was then normalized and plotted.

### Quantification of LUCIFERASE (LUC) activity in *Arabidopsis* roots

*DR5::LUC* expressing 7 DAG *Arabidopsis* seedlings were transferred to ammonium or nitrate-containing agar plates, and roots (40 roots per treatment per time point) were collected after 1 and 6 HAT and snap frozen in liquid nitrogen. Frozen root tissue was extracted in Reporter Lysis Buffer (Promega), and LUC activity was measured with the Luciferase Assay Reagent (Promega) in a multiwell plate in a Biotek Synergy H1 platereader.

### Measurements of basipetal (shootward) auxin transport in *Arabidopsis* roots

The shootward transport assay of [3H]-IAA in *Arabidopsis* roots was performed according to a previous report (Lewis & Muday, 2009), with a few modifications. 7 DAG Col-0 or *eir1-4* seedlings were transferred to ammonium, nitrate, or MS (Murashige Skoog Basal Medium) medium with 15 seedlings as one biological replicate, and three replicates per treatment. The [3H]-IAA (PerkinElmer, ART-0340) droplets were prepared in MS medium with 1.25% agar and 500 mM [3H]-IAA (1.45 ml in 10 ml) and were carefully placed on the root meristem (at the very end of the roots). After incubation for 6 h in the dark, the part of the root which was covered by the droplet was cut, and the remaining root parts were collected and ground completely in liquid nitrogen and homogenized in 1 ml scintillation solution (PerkinElmer, 6013199). The samples were incubated overnight to allow the radioactivity to evenly diffuse into the whole volume of the scintillation cocktail. Finally, the radioactivity

was measured with a scintillation counter (Hidex 300XL), with each sample counted for 100 s, three times. Three samples with only the scintillation solution were used as background controls. As an additional background control, another batch of samples were prepared the same way as described above except [3H]-IAA containing droplets were placed not on the root meristem but next to the seedlings. Data shown on the figure were calculated against the background.

## GUS (β-Glucuronidase) staining

*CycB1::GUS* expression was analyzed in seedling roots 7 DAG, 12 HAT to ammonium or nitrate-containing media. Seedlings were incubated for 2 h in 37°C in staining buffer containing 1 mM ferricyanide, 150 mM sodium phosphate buffer (pH 7), and 1 mg/ml of X-Gluc dissolved in DMSO. Seedlings were cleared using subsequent incubation at room temperature in a series of ethanol dilutions from 60 to 10% then mounted on slides with 5% ethanol-50% glycerol mounting solution. The pattern of the GUS histochemical staining was analyzed by an Olympus BX53 microscope and Olympus DP26 digital camera, controlled by cellSense Entry software.

## RT–qPCR analysis

Total RNA was extracted from excised 7 DAG roots 1, 6, and 48 HAT to ammonium or nitrate amended plates using RNeasy® Plant Mini kit (QIAGEN) according to the manufacturer's protocol. 1 μg of RNA was used to synthesize cDNA using iScriptTM cDNA synthesis kit (Bio-Rad). The analysis was carried out on a LightCycler 480 II (SW1.5.1 Version; Roche Diagnostics) with the SYBR Green I Master kit (Roche Diagnostics) according to the manufacturer's instructions. All PCR reactions were carried out with biological and technical triplicates. Expression levels of target genes were quantified by specific primers that were designed using Quant Prime (Arvidsson *et al*, 2008) and validated by performing primer efficiency for each primers pair. The levels of expression of each gene were first measured relative to *AT4G05320* (*UBQ10*) and then to respective mock treatment.

| Gene | Transcript Identifier | Primer FW | Primer REV |
|------|----------------------|-----------|------------|
| AT4G05320 | UBQ10 | CACACTCCAC TTGGTCTTGC | TGGTCTTTCC GGTGAGAG TCTTCA |
| PIN2 | AT5G57090.1 | TCACGACAACC TCGCTACTAAAGC | TGCCCATGT AAGGTGAC TTTCCC |
| ANR1 | AT2G14210.1 | AAGAGGAGCA GCATCAACTTCTG | TCCTCTCC CACTAGTTT CCTGTG |

## Reproducibility and statistics

The number of independent repetitions of experiments, as well as exact sample sizes, is described in the figure legends. Statistical analysis (*t*-test and ANOVA) were performed using the software Origin (v2018). Statistical significance was tested as described in the figure legends.

For the regression analysis in supplementary document 1, Col-0 cell length measurements were analyzed together with associated categorical variables represented by plant sample of origin ($n = 18$), tissue ($n = 2$, i.e., epidermis and cortex), cell position ($n = 20$) and treatment ($n = 2$, i.e., $NO_3^-$ and $NH_4^+$). The importance of the variables was initially accessed via Random Forest analysis in R (v 1.2.5033). A machine learning training was conducted with the caret R package (Kuhn *et al*, 2020) for tuning the Random Forest and the best mtry parameter was selected according to root mean square error (RMSE) and R-squared (R2) measures in R. Data distribution, skewness, and kurtosis were checked with the fitdistrplus R package (Delignette-Muller & Dutang, 2015), and Gamma distribution was chosen for setting up a regression analysis based on generalized linear models (GLMs). Besides the main effects of the variables, several models were virtually possible when the interactions between some or all the variables were considered. First, a simple model including only main effects was generated and residual vs. fitted values evaluated prior to analysis of deviance. Second, a model including main effects and all possible interactions between variables was built. The analysis of the interactions of the fit model was carried out with the phia R package (Rosario-Martinez *et al*, 2015), showing a possible but not strong interaction between tissue and treatment factors. This insight was used to generate a third model. The performance of the second and third model was then compared by repeated k-fold cross-validation with the caret R package, and the second model was selected according to RMSE and R2 measures. After analysis of deviance, post hoc pairwise comparisons were conducted with estimated marginal means (EMMs) using the emmeans R package (Lenth *et al*, 2020) (Appendix Fig S9A and B). A recursive partitioning analysis was performed, and a decision tree was generated with the partykit R package (Hothorn & Zeileis, 2015) to confirm the results showed by regression analysis and to visualize the role of different variables on cell length distribution (Appendix Fig S9C).

## Computational methods

### Visualization of model predictions

The computer simulation representing the dynamic auxin flow through the root tissues was created using the version of VV (Vertex–Vertex) programming language and in the L-system-based modeling software L-studio (Karwowski & Prusinkiewicz, 2004). The model simulates a cross section of the plant root focusing on the cortical and epidermal tissues. Plant cells are visualized as four-sided polygons representing the cell walls. For the sake of simplicity, cell membranes and the extracellular space shared by adjacent cells are not rendered. Only the first ~ 20 cells (counting from the QC) are visualized, mirroring the available experimental measurements. Meristematic and elongating cells are distinguished with different cell wall coloring, blue for meristem and yellow for elongation zone, respectively. Auxin is represented as filled green circles inside each cell, the radius of the circle proportional to the size of the cell indicates the amount of auxin present in that cell. PIN2 protein localization on the PM is represented as red dots close to the cell walls; PIN2 can be apical (shootward), basal (rootward), or lateral (outer). Despite being taken into account for mathematical calculations, cytoplasmic accumulation of PIN2 is not shown in the model visualizations. Our model enables dynamic simulation of root

growth, elongation, and auxin flow through the root apex. Individual cells grow, elongate, and consequently divide. Auxin is pumped across cell walls through the ATP-dependent action of PIN2 proteins on the cell membrane. Auxin that reaches the outer limit of the tissues is simply removed from the system. PIN2 is expressed, trafficked, and degraded according to the model rules described in the following sections.

### Mathematical model description

The model assumes that the epidermis contributes to an active passage auxin into deeper tissues. Two main sources of auxin into the epidermis were considered:

1   The cell that is closest to the QC, which is known to be a main source of auxin production (Stepanova *et al*, 2008).
2   The lateral root cap, which due to its structural conformation force the influx of auxin into the initial cells of the epidermis (Xuan *et al*, 2016).

The ordinary differential equation describing auxin dynamic in a single cell $i$ is as follows:

$$\frac{dA_i}{dt} = (s_1 \cdot i < z + s_2 \cdot i \geq z) + \sum_{j=1}^{n} k_a(A_j PIN_j - A_i PIN_i) - d_a A_i$$

where $s_1$ and $s_2$ denote the two auxin sources into the epidermis), while $z$ indicates the cell location of the LRC-derived auxin influx (cell number 20 from the QC). $k_a$ represents the rate of active auxin transport between cells via PIN2. The exchange of auxin occurs for each cell $j$ connected to cell $i$. General processes of auxin degradation like conjugation and oxidation are summarized by a single degradation rate, $d_a$.

PIN2 is the only auxin efflux carries considered in this model. High auxin concentrations lead to an increased degradation of PIN proteins (Kleine-Vehn *et al*, 2008; Kleine-Vehn & Friml, 2008). We modeled the effect of auxin on cytoplasmic PIN2 inside cell $i$ by approximating functional forms, in what follow:

$$\frac{dPIN_{ci}}{dt} = m_p - d_p PIN_{ci} \cdot \left(1 + \frac{A_i}{q_p}\right)$$

The expression parameter $m_p$, indicates the basal rate of PIN2 protein synthesis. PIN2 degradation is modeled over a constant rate of degradation $d_p$, which increases linearly according to auxin levels by $q_p$.

PIN2 trafficking to the apical/basal membranes is modeled as follows:

$$\frac{dPIN_{mi}}{dt} = PIN_{ci}(1 - l_n)((tr_n + tr_{wn}) \cdot (1 - tr_n + tr_{wn})) \cdot \text{Logistic}(tr_a A_i + tr_i i)$$

The amount of PIN2 on the membrane $m$ of cell $i$ is regulated by the basal trafficking rates on $NO_3^-$ ($tr_n$) or $NH_4^+$ ($tr_{wn}$), which in turn is allowed to saturate to zero or to the maximum rate depending on the level of auxin and the distance from the QC, according to logistic coefficients $tr_a$ and $tr_i$, respectively. $l_n$ represents the percentage of PIN2 that is redirected to the lateral membranes, depending on nitrate levels. In this model, nitrate

level is represented as a binary variable: $NO_3^-$ for nitrate supplement and 0 for $NH_4^+$ supplement.

Cell division is regulated through a hypothetical division factor as proposed in a previous study (Mironova *et al*, 2010). The concentration of division factor in a single cell $i$ is describes as:

$$\frac{dDIV_i}{dt} = k_{v0} \cdot \frac{\left(k_{v1} \cdot \frac{A_i}{\max A}\right) + \left(\frac{len_i}{\max L}\right)}{1 + e^{(i \cdot t_v)}} - DIV_i \cdot k_{v2} \frac{1 + \left(\frac{A_i}{k_{v3}}\right)^{h_1}}{1 + \left(\frac{A_i}{k_{v4}}\right)^{h_2}}$$

where $k_{v0}$ denotes the maximal synthesis rate of division factor; $len_i$ and $maxL$ the length of cell $i$ and the maximum cell length achievable, respectively; $t_v$ is the tolerance factor restricting the location in the meristem where division takes place; $k_{v1}$ is the level of auxin-dependent division factor activation. The right part of the formula describes the hypothesized process of division factor degradation, where $k_{v2}$ is the degradation rate of the division factor; $k_{v3}$ and $k_{v4}$ are the level of auxin-dependent division factor activation and saturation, respectively; $h_1$ and $h_2$ are hill's coefficient.

Cell growth is an auxin-dependent mechanism, and cell entrance in the elongation phase is triggered by an auxin concentration threshold. Both in the meristem and the elongation zone cell growth is defined as follow:

$$\frac{dL_i}{dt} = k_l \cdot \frac{A_i}{A_i + 1} L_i \cdot \left(1 - \frac{len_i}{m_l}\right)$$

where $k_l$ indicates the cell elongation rate (depending on whether the cell in the meristem or in the elongation zone), $len_i$ the cell length, and $m_l$ the maximum length the cell can achieve (depending whether the cell in the meristem or in the elongation zone).

### Statistical inference and parameters estimation

Data analysis and plotting were performed using the R language environment for statistical computing (R Core Team, 2017) and the plotting package ggplot (Ginestet, 2011). Parameters estimation of the previously described models was carried out with the RStan (Carpenter *et al*, 2017) and brms (Bürkner, 2017) packages, which implement a modified version of Hamiltonian Monte Carlo sampling algorithm to approximate the parameters posterior distribution. Model comparison was performed using the loo package (Vehtari *et al*, 2017) to carry out Pareto smoothed importance-sampling leave-one-out cross-validation (PSIS-LOO) for posterior predictive performance estimation.

### Auxin source implementation and testing

To test auxin source impact on the model predictions, we considered four possible scenarios (Appendix Fig S7F):

1   Model A: A naive model assuming a uniform source of auxin along the epidermis (uniform source).
2   Model B: The current model that consider two separate sources from the QC and the LRC (LRC source).
3   Model C: A highly complex model that assume input auxin source modeled as a versatile spline (spline source across epidermis).
4   Model D: A more complex but less realistic model allowing for different independent input of auxin for each epidermis cell (multiple point source).

To identify the best model, we tested Models A–D against experimental measurements and generate the information criteria based on the expected log probability density (Appendix Fig S7F). A lowest information criterion was found for Model B as indicated the posterior predictive performance, thereby Model B was used for the further study. We decided to exclude the existence of a significant influx of auxin into the cortical cells; this was backed by previous researches which suggested that at high auxin levels endodermal cells have the tendency to lateralize toward the internal tissues and not toward the cortex (Sauer *et al*, 2006).

### Parameters values used in the model

Parameters values used in the model are listed below with their estimated mean and lower/upper 95% credible intervals.

| Parameter | Mean | l-95% CI | u-95% CI | Reference (when not estimated from data) |
|---|---|---|---|---|
| $d_a / k_a$ | 0.018 | 0.018 | 0.018 | Mironova *et al* (2010) |
| $s_1$ | 8.36 | 0.44 | 21.54 | – |
| $s_2$ | 22.53 | 9.13 | 36.46 | – |
| $z$ | 10.6 | 8.39 | 12.02 | – |
| $m_p$ | 30.49 | 16.69 | 46.00 | – |
| $d_p$ | 0.065 | 0.051 | 0.079 | – |
| $q_p$ | 100 | 100 | 100 | Wabnik *et al* (2010) |
| $l_n$ | 0.60 | 0.51 | 0.69 | – |
| $tr_n$ | 0.246 | 0.226 | 0.267 | – |
| $tr_{wn}$ | 0.13 | 0.11 | 0.15 | – |
| $tr_a$ | −0.05 | −0.10 | −0.00 | – |
| $tr_i$ | 0.30 | 0.25 | 0.34 | – |
| $k_{v0}$ | 1.5 | 1.5 | 1.5 | Mironova *et al* (2010) |
| $k_{v1}$ | 20 | 20 | 20 | Mironova *et al* (2010) |
| $k_{v2}$ | 0.3 | 0.3 | 0.3 | Mironova *et al* (2010) |
| $k_{v3}$ | 3.5 | 3.5 | 3.5 | Mironova *et al* (2010) |
| $k_{v4}$ | 0.5 | 0.5 | 0.5 | Mironova *et al* (2010) |
| $t_v$ | 0.1 | 0.1 | 0.1 | Mironova *et al* (2010) |
| $h_1$ | 2 | 2 | 2 | Mironova *et al* (2010) |
| $h_2$ | 3 | 3 | 3 | Mironova *et al* (2010) |
| $k_l$ | 0.3 | 0.2 | 0.4 | Yang *et al* (2017) |
| $m_l$ | 200 | 150 | 250 | Yang *et al* (2017) |

## Data availability

This study includes no data deposited in external repositories.

**Expanded View** for this article is available online.

## Acknowledgements

We acknowledge Gergely Molnár for critical reading of the manuscript, Alexander Johnson for language editing and Yulija Salanenka for technical assistance. Work in the Benková laboratory was supported by the Austrian Science Fund (FWF01_I1774S) to KÖ, RA and EB. Work in the Benkova laboratory was supported by the Austrian Science Fund (FWF01_I1774S) to KO, RA and EB and by the DOC Fellowship Programme of the Austrian Academy of Sciences (25008) to C.A. Work in the Wabnik laboratory was supported by the Programa de Atracción de Talento 2017 (Comunidad de Madrid, 2017-T1/BIO-5654 to K.W.), Severo Ochoa Programme for Centres of Excellence in R&D from the Agencia Estatal de Investigación of Spain (grant SEV-2016-0672 (2017-2021) to K.W. via the CBGP) and Programa Estatal de Generación del Conocimiento y Fortalecimiento Científico y Tecnológico del Sistema de I+D+I 2019 (PGC2018-093387-A-I00) from MICIU (to K.W.). M.M. was supported by a postdoctoral contract associated to SEV-2016-0672. We acknowledge the Bioimaging Facility in IST-Austria and the Advanced Microscopy Facility of the Vienna BioCenter Core Facilities, member of the Vienna BioCenter Austria, for use of the OMX v4 3D SIM microscope. AJ was supported by the Austrian Science Fund (FWF): I03630 to J.F.

## Author contributions

KÖ and EBe conceived the project; KÖ performed most of the experiments; MM and KW designed the computer model; MM implemented the model and performed computer model simulations; AV, JO, and RAG shared unpublished material; AJ performed the 3D-SIM experiment; RA performed qPCR and GUS-staining experiments; LA performed the regression analysis; JCM contributed to the multiphoton microscopy imaging; YZ conducted protein sequence alignments; ST generated the DII and mDII lines in the *eir1-4* background; CC, EBo, and AG designed and performed some of the pre-pilot experiments; CA assisted KÖ in multiple experiments; JF provided infrastructure, resources, and scientific supervision for AJ, YZ, and ST The manuscript was written by KÖ, KW, and EBe.

## Conflict of interest

The authors declare that they have no conflict of interest.

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
