## [Review Process File · The EMBO Journal]

Modulation of plant root growth by nitrogen source-defined regulation of polar auxin transport

Krisztina Otvos, Marco Marconi, Andrea Vega, Jose O' Brien, Alexander Johnson, Rashed Abualia, Livio Antonielli, Juan Carlos Montesinos, Yuzhou Zhang, Shutang Tan, Candela Cuesta, Christina Artner, Eleonore Bouguyon, Alain Gojon, Jiri Friml, Rodrigo Gutiérrez, Krzysztof Wabnik, and Eva Benková

DOI: [10.15252/embj.2020106862](https://doi.org/10.15252/embj.2020106862)

Corresponding authors: Eva Benková (Eva.BENKOVA@ist.ac.at) , Krzysztof Wabnik (k.wabnik@upm.es)

Review Timeline:	Transfer from Review Commons:	21st Sep 20
	Editorial Decision:	28th Sep 20
	Revision Received:	16th Oct 20
	Editorial Decision:	23rd Oct 20
	Revision Received:	27th Oct 20
	Accepted:	6th Nov 20

Editor: Ieva Gailite

Transaction Report:

This manuscript was transferred to The EMBO Journal following peer review at Review Commons.

Thank you for submitting your manuscript for consideration by the EMBO Journal. I sincerely apologise for the delay in the assessment of your study due to the high number of submissions our office is experiencing at the moment. I have now read your manuscript, the reviewer comments and your response to them. Based on our editorial assessment and the referees' positive evaluations, I would like to invite you to submit a revised version of the manuscript.

I understand that the manuscript you have submitted to us has already been revised in response to the reviewer comments, but the invitation to revise is a formal and technical requirement from our side to be able to resume the review process. Moreover, we routinely perform an initial quality control on all revised manuscripts before re-review, for which we require the files to be in a specific format. Please revise your manuscript according to the instructions that follow below. I would also like to suggest including the data generated in response to Reviewer #2 in the manuscript, as these findings would be a valuable addition to the study.

Please feel free to contact me if you have any further questions regarding the revision. Thank you for the opportunity to consider your work for publication. I look forward to receiving your revised manuscript.

Review #1

1. How much time do you estimate the authors will need to complete the suggested revisions:

Estimated time to Complete Revisions (Required)

(Decision Recommendation)

Less than 1 month

2. Evidence, reproducibility and clarity:

Evidence, reproducibility and clarity (Required)

****Summary**** This work is a nice demonstration of cutting-edge live imaging of Arabidopsis roots grown on different nitrogen sources, and it links root response to nitrogen source to auxin distribution by altered PIN2 function. It presents evidence that PIN2 phosphorylation on an uncharacterized site in part regulates this response. The authors code a mathematical model that recapitulates the major properties of the root response to nitrate observed in the paper. ****Major comments:**** I have no major comments. ****Minor comments:**** I think the message of the paper would be stronger if a few points were made more clear. First, I had to re-read sections of the paper multiple times to grasp the elongation/division asynchrony phenotype. I think this is quite an interesting finding but I think it would help to describe it more directly. As an example, the abstract states: "We show that growth, i.e. tissue-specific cell division and elongation rates are fine-tuned by modulating auxin flux within and between tissues." I find this phrase leaves the reader still highly uncertain of what the basic phenotype is. I think it would help the reader to more precisely describe, upfront, the finding- ie, that growth on ammonium produced asynchrony in growth/division in root cell layers that was rescued by transfer to nitrate. Similarly, the direction of regulation by phosphorylation of S439 is a little hard to keep in mind. There are phrases like this: "Altogether, these results suggest that phosphorylation status of PIN2 on S439 account for fine-tuning of PIN2 trafficking towards the PM and polarity establishment under varying N sources." and "The functional characterization of PIN2 and its phosphor variants suggests that N source dependent regulation of PIN2 phosphorylation status has a direct impact on the flexible adjustment of PIN2 membrane localization and polarity, and thereby adaptation of root growth to varying forms of N supply" Again, it leaves the reader having to process a lot of information about what phosphorylation or dephosphorylation of this site is likely to do. It would be helpful for reader comprehension to state a model- that phosphorylation at this site leads to reduced lateralization when plants are on ammonium, and dephosphorylation promotes lateralization on nitrate, etc. Also, the above phrase ("The functional characterization of PIN2...") was repeated verbatim in the introduction and discussion.

3. Significance:

Significance (Required)

This work adds a new molecular link connecting regulation of root growth, at the cellular level, to whole-plant growth dynamics and nitrogen source availability. It will be of interest to the very large number of researchers concerned with polar auxin transport, as well as plant nutrient researchers, and others. The authors do a good job of being cautious in interpretation by not making unfounded assertions, and the data presentation is detailed and strong. My area of expertise is in plant cell/hormone signaling, so I am unable to assess the adequacy of the model, although it appears to be logical. In my mind the most important limitation of the study is the artificial experimental system where plants are grown on ammonium as a sole nitrogen source and abruptly transferred to nitrate. As the authors note, nitrate is the predominant nitrogen source in aerated soil where Arabidopsis typically grows. I think it is fair to question if the relatively lower abundance of PIN2, reduced PIN2 lateralization, and asynchronous cortex/epidermal division and elongation in plants grown on ammonium represent a specific, evolved response to this nitrogen source, or if the preference for nitrate of Arabidopsis means the phenotype on ammonium is a consequence of a more general stress condition. There is a difference between being able to say positively that the plant "tunes" growth based on nitrogen source (which sounds very purposeful), versus only being able to say that Arabidopsis did not evolve to subsist on only ammonium and it experiences stress when doing so. This is not a fatal flaw. I think the results in the paper are meaningful and provide interesting future directions for the study of PIN2 regulation, and of S439 in particular, during environmental adaptation, but I am not convinced that this is a specific mechanism evolved in nature to help plants cope with different nitrogen sources. This limits the generalizability of the findings at the present time. REFEREE'S CROSS-COMMENTING: The reviews seem consistent that this is a well-presented and interesting paper. They both agree that there are only a small number of changes required for publication.

Review #2

1. How much time do you estimate the authors will need to complete the suggested revisions:

Estimated time to Complete Revisions (Required)

(Decision Recommendation)

Less than 1 month

2. Evidence, reproducibility and clarity:

Evidence, reproducibility and clarity (Required)

****Summary:**** Particularly impressive is how this work brings together different fields of study - using high-resolution microscopy in combination with the study of growth dynamics and N-kinetics. It then goes on to implicate both auxin and auxin transport via subcellular trafficking of its transporters in this, to help work out if altered auxin flow is a cause or effect - occurring early or later - during the adaptation process. The manuscript is well presented and the figures prepared to a very high standard. I would recommend to the journal to make these figures as large as possible, to enable the fine detail to be clearly seen. There are a lot of supplemental figures but these have useful information to help back up the conclusions. In addition, the videos provide added value to the article. I thought that the decision tree figure was engaging, whilst also offering a robust way to categorise and analyse the data. ****Major comments:**** Since variable auxin level is well-known to be important (Model B, not Model A) I would like to see a greater variety of models discussed in the main text. What about taking into account variable flow over day/night, for example? Could such changes affect how plants adapt to new N sources at night-time compared to during the day? Some of the figures mention a certain number of roots (e.g. Figure 1) and others mention a certain number with independent experiments (e.g. Figure 2) - please can you clarify if some experiments had true biological repeats and others not? They are all biological repeats in a sense since each measurement comes from a different seedling, but this was rather opaque. Supplementary Figure 10 is ok but it would be more useful in the form of a more realistic root diagram - showing the cell types and locations of activity relative to basipetal auxin flow. ****Very minor comments:**** Line 496 error: molecular mechanisms describing how

3. Significance:

Significance (Required)

This work uses a range of physiological, molecular and imaging methods to investigate how plants adapt their growth to differing nitrogen levels and nitrogen forms. The authors find that cell elongation, per layer, changes in response to different environmental conditions, and is also distinct depending on the form of nitrogen. We already know that the form of nitrogen can be sensed differently (or at least there are different responses to it) depending on cell type, and this work nicely develops those ideas. It was sensible to focus on 'growth' and thus profile the epidermis and cortex here. However, this does set up the possibility of studying changes in the pericycle and thus lateral root development-related adaptations, which would be great to see in the future. This manuscript would be of wide interest to anyone interested in how multicellular systems respond to change and partition functions. REFEREE'S CROSS-COMMENTING: I think our comments are well in agreement and suggests that a small amount of redrafting will help the authors to present a clear message.

Manuscript EMBOJ-2020-106862**Response to Reviewers**

Dear Editor,

Thank you for giving us the opportunity to submit a revised version of the manuscript “Modulation of root growth by nutrient-defined regulation of polar auxin transport” for publication in The EMBO Journal. We appreciate the time and efforts that you and the reviewers dedicated to providing feedback on our manuscript and we are grateful for the insightful comments, which we found very helpful to improve the presentation and the main text of the manuscript.

We have incorporated all the suggestions made by the reviewers. Those changes are highlighted within the manuscript. Please see below a point-by-point response to the reviewers’ comments and concerns. All line number refer to the revised manuscript file with tracked changes.

Reviewers' Comments to the Authors:

Reviewer #1 (Evidence, reproducibility and clarity (Required)):

****Summary****

This work is a nice demonstration of cutting-edge live imaging of Arabidopsis roots grown on different nitrogen sources, and it links root response to nitrogen source to auxin distribution by altered PIN2 function. It presents evidence that PIN2 phosphorylation on an uncharacterized site in part regulates this response. The authors code a mathematical model that recapitulates the major properties of the root response to nitrate observed in the paper.

****Major comments:****

I have no major comments.

****Minor comments:****

I think the message of the paper would be stronger if a few points were made more clear. First, I had to re-read sections of the paper multiple times to grasp the elongation/division asynchrony phenotype. I think this is quite an interesting finding but I think it would help to describe it more directly. As an example, the abstract states:

"We show that growth, i.e. tissue-specific cell division and elongation rates are fine-tuned by modulating auxin flux within and between tissues."

I find this phrase leaves the reader still highly uncertain of what the basic phenotype is. I think it would help the reader to more precisely describe, upfront, the finding- ie, that growth on ammonium produced asynchrony in growth/division in root cell layers that was rescued by transfer to nitrate. Similarly, the direction of regulation by phosphorylation of S439 is a little hard to keep in mind. There are phrases like this:

"Altogether, these results suggest that phosphorylation status of PIN2 on S439 account for fine-tuning of PIN2 trafficking towards the PM and polarity establishment under varying N sources." and "The functional characterization of PIN2 and its phosphor variants suggests that N source dependent regulation of PIN2 phosphorylation status has a direct impact on the flexible adjustment of PIN2 membrane localization and polarity, and thereby adaptation of root growth to varying forms of N supply"

Again, it leaves the reader having to process a lot of information about what phosphorylation or dephosphorylation of this site is likely to do. It would be helpful for reader comprehension to state a model- that phosphorylation at this site leads to reduced lateralization when plants are on ammonium, and dephosphorylation promotes lateralization on nitrate, etc.

Also, the above phrase ("The functional characterization of PIN2...") was repeated verbatim in the introduction and discussion.

Author response: Thank you for pointing this out. We agree that presentation of specific aspects of PIN2 mediated transport and root growth adaptation to either ammonium or nitrate was not sufficiently clear. In the revised manuscript we paid attention to more clear and

straightforward description of the respective findings (lines (45-56, 72-133, 158-163, 220-221, 252-269, 289-291, 376-377, 420-422, 426-440, 450-458, 495-543, 553-554, 557-569, 593-613, 625-633, 648-650, 666-682 in the “EMBOJ-2020-106862_Tracked changes_manuscript text.pdf” file).

Reviewer #1 (Significance (Required)):

This work adds a new molecular link connecting regulation of root growth, at the cellular level, to whole-plant growth dynamics and nitrogen source availability. It will be of interest to the very large number of researchers concerned with polar auxin transport, as well as plant nutrient researchers, and others. The authors do a good job of being cautious in interpretation by not making unfounded assertions, and the data presentation is detailed and strong. My area of expertise is in plant cell/hormone signaling, so I am unable to assess the adequacy of the model, although it appears to be logical.

In my mind the most important limitation of the study is the artificial experimental system where plants are grown on ammonium as a sole nitrogen source and abruptly transferred to nitrate. As the authors note, nitrate is the predominant nitrogen source in aerated soil where *Arabidopsis* typically grows. I think it is fair to question if the relatively lower abundance of PIN2, reduced PIN2 lateralization, and asynchronous cortex/epidermal division and elongation in plants grown on ammonium represent a specific, evolved response to this nitrogen source, or if the preference for nitrate of *Arabidopsis* means the phenotype on ammonium is a consequence of a more general stress condition. There is a difference between being able to say positively that the plant "tunes" growth based on nitrogen source (which sounds very purposeful), versus only being able to say that *Arabidopsis* did not evolve to subsist on only ammonium and it experiences stress when doing so. This is not a fatal flaw. I think the results in the paper are meaningful and provide interesting future directions for the study of PIN2 regulation, and of S439 in particular, during environmental adaptation, but I am not convinced that this is a specific mechanism evolved in nature to help plants cope with different nitrogen sources. This limits the generalizability of the findings at the present time.

Authors response: We agree with the reviewer. We also asked whether the developmental and growth adaptation of roots to either ammonium or nitrate represent a specific, nitrogen source

dependent response, or it might be the consequence of a more general stress condition, caused by the experimental setup. To address this question we performed three sets of experiments (please see Fig. EV5 and lines 479-512 in the revised manuscript).

We hope that in the light of the presented experiments the reviewer is more convinced about the nitrogen source specificity of our findings.

REFEREE'S CROSS-COMMENTING:

The reviews seem consistent that this is a well-presented and interesting paper. They both agree that there are only a small number of changes required for publication.

Reviewer #2 (Evidence, reproducibility and clarity (Required)):

****Summary:****

Particularly impressive is how this work brings together different fields of study - using high-resolution microscopy in combination with the study of growth dynamics and N-kinetics. It then goes on to implicate both auxin and auxin transport via subcellular trafficking of its transporters in this, to help work out if altered auxin flow is a cause or effect - occurring early or later - during the adaptation process.

The manuscript is well presented and the figures prepared to a very high standard. I would recommend to the journal to make these figures as large as possible, to enable the fine detail to be clearly seen. There are a lot of supplemental figures but these have useful information to help back up the conclusions. In addition, the videos provide added value to the article. I thought that the decision tree figure was engaging, whilst also offering a robust way to categorise and analyse the data.

****Major comments:****

Since variable auxin level is well-known to be important (Model B, not Model A) I would have liked to see a greater variety of models discussed in the main text. What about taking into account variable flow over day/night, for example? Could such changes affect how plants adapt to new N sources at night-time compared to during the day?

Author's response: Thank you for pointing out the day and night rhythmicity. The reviewer is correct and we are aware of the studies showing that the circadian clock regulates auxin signaling and responses¹. However, as we show in Figure 1a, although day-night cycling affects root growth, we could not observe any measurable nitrogen source-specific component in this effect. Therefore, in the presented models, we think the circadian effect is so minor that makes it negligible.

Some of the figures mention a certain number of roots (e.g. Figure 1) and others mention a certain number with independent experiments (e.g. Figure 2) - please can you clarify if some experiments had true biological repeats and others not? They are all biological repeats in a sense since each measurement comes from a different seedling, but this was rather opaque. Supplementary Figure 10 is ok but it would be more useful in the form of a more realistic root diagram - showing the cell types and locations of activity relative to basipetal auxin flow.

Authors response:

We apologize for impreciseness regarding “biological repeats”. All of the presented experiments have biological repetitions. We mean biological replicates on “independent experiments”. For example in Figure 1a “three independent experiments (each consisting of 3 roots per treatment)” means that the presented data represents the geomean of 9 roots from 3 biological repetitions. We would like to mention here that due to the experimental setup of the real-time imaging we are limited in the numbers of the imaged roots per experiment. In Figure 2 “Data are derived from 5 roots per condition from three independent experiments” means we grouped measurements from 3 biological replicates. We corrected the phrase ‘independent experiments’ to ‘biological replicates’ in the manuscript.

Supplemental Figure 10 was corrected, please see as Appendix Fig. S8.

****Very minor comments:****

Line 496 error: molecular mechanisms describing how

Authors response: Thank you for noticing that, it has been corrected.

Reviewer #2 (Significance (Required)):

This work uses a range of physiological, molecular and imaging methods to investigate how plants adapt their growth to differing nitrogen levels and nitrogen forms. The authors find that cell elongation, per layer, changes in response to different environmental conditions, and is also distinct depending on the form of nitrogen. We already know that the form of nitrogen can be sensed differently (or at least there are different responses to it) depending on cell type, and this work nicely develops those ideas.

It was sensible to focus on 'growth' and thus profile the epidermis and cortex here. However, this does set up the possibility of studying changes in the pericycle and thus lateral root development-related adaptations, which would be great to see in the future.

This manuscript would be of wide interest to anyone interested in how multicellular systems respond to change and partition functions.

REFEREE'S CROSS-COMMENTING:

I think our comments are well in agreement and suggests that a small amount of redrafting will help the authors to present a clear message.

Bibliography

1. The Circadian Clock Regulates Auxin Signaling and Responses in Arabidopsis.
[doi:10.1371/journal.pbio.0050222.](https://doi.org/10.1371/journal.pbio.0050222)

Thank you for submitting the revised version of your Review Commons manuscript. To my assessment, reviewer comments are sufficiently addressed, and there now remain only a few editorial issues that have to be solved before I can extend formal acceptance of the manuscript.

The authors performed the requested changes.

Editor accepted the manuscript.

Corresponding Author Name: Eva Benkova

Journal Submitted to: The EMBO Journal

Manuscript Number: EMBOJ-2020-106862